# *TET2* lesions enhance the aggressiveness of *CEBPA*-mutant acute myeloid leukemia by rebalancing *GATA2* expression

Elizabeth Heyes[1,10], Anna S. Wilhelmson [2,3,4,10], Anne Wenzel [2,3,4], Gabriele Manhart[1], Thomas Eder [1], Mikkel B. Schuster[2,3,4], Edwin Rzepa[1], Sachin Pundhir[2,3,4], Teresa D'Altri[2,3,4], Anne-Katrine Frank [2,3,4], Coline Gentil[2,3,4], Jakob Woessmann [5], Erwin M. Schoof [5], Manja Meggendorfer [6], Jürg Schwaller [7], Torsten Haferlach [6], Florian Grebien [1,8,11] ✉ & Bo T. Porse [2,3,4,9,11] ✉

The myeloid transcription factor CEBPA is recurrently biallelically mutated (i.e., double mutated; *CEBPA*[DM]) in acute myeloid leukemia (AML) with a combination of hypermorphic N-terminal mutations (*CEBPA*[NT]), promoting expression of the leukemia-associated p30 isoform, and amorphic C-terminal mutations. The most frequently co-mutated genes in *CEBPA*[DM] AML are *GATA2* and *TET2*, however the molecular mechanisms underlying this co-mutational spectrum are incomplete. By combining transcriptomic and epigenomic analyses of *CEBPA-TET2* co-mutated patients with models thereof, we identify *GATA2* as a conserved target of the *CEBPA-TET2* mutational axis, providing a rationale for the mutational spectra in *CEBPA*[DM] AML. Elevated CEBPA levels, driven by *CEBPA*[NT], mediate recruitment of TET2 to the *Gata2* distal hematopoietic enhancer thereby increasing *Gata2* expression. Concurrent loss of TET2 in *CEBPA*[DM] AML induces a competitive advantage by increasing *Gata2* promoter methylation, thereby rebalancing GATA2 levels. Of clinical relevance, demethylating treatment of *Cebpa-Tet2* co-mutated AML restores *Gata2* levels and prolongs disease latency.

Acute myeloid leukemia (AML) is characterized by genetic alterations affecting the proliferation and/or differentiation of hematopoietic stem or progenitor cells (HSPCs). Thereby, the expansion of immature myeloid precursors, at the expense of normal hematopoiesis, ultimately leads to bone marrow (BM) failure if left untreated. Recent sequencing efforts have identified numerous recurrent mutations in AML and revealed patterns of mutational co-segregation, suggesting

that synergism between certain lesions drives leukemogenesis[1]. While we now recognize these patterns, the mechanistic basis for context-specific positive or negative selection of certain lesions remains to be elucidated in most cases.

CCAAT enhancer binding protein alpha (CEBPA) is a hematopoietic lineage-specific transcription factor that binds and primes genes for myeloid development and is required for differentiation and

[1]University of Veterinary Medicine, Institute of Medical Biochemistry, Vienna, Austria. [2]The Finsen Laboratory, Copenhagen University Hospital - Rigshospitalet, Copenhagen, Denmark. [3]Biotech Research and Innovation Centre (BRIC), Faculty of Health Sciences, University of Copenhagen, Copenhagen, Denmark. [4]Danish Stem Cell Center (DanStem), Faculty of Health Sciences, University of Copenhagen, Copenhagen, Denmark. [5]Department of Biotechnology and Biomedicine, Technical University of Denmark, Lyngby, Denmark. [6]MLL Munich Leukemia Laboratory, Munich, Germany. [7]Department of Biomedicine, University Children's Hospital Basel, Basel, Switzerland. [8]St. Anna Children's Cancer Research Institute (CCRI), Vienna, Austria. [9]Department of Clinical Medicine, University of Copenhagen, Copenhagen, Denmark. [10]These authors contributed equally: Elizabeth Heyes, Anna S. Wilhelmson. [11]These authors jointly supervised this work: Florian Grebien, Bo T. Porse. ✉e-mail: Florian.Grebien@vetmeduni.ac.at; bo.porse@finsenlab.dk

maturation of granulocytes[2]. The gene encoding CEBPA is biallelically mutated (i.e., double mutated; *CEBPA*[DM]) in 3–15% of de novo AML patients[3–9]. *CEBPA*[DM] patients harbor either biallelic N-terminal mutations or a combination of a monoallelic N-terminal mutation together with a C-terminal mutation in the other allele. Whereas N-terminal CEBPA (*CEBPA*[NT]) lesions promote the expression of the truncated p30 isoform, C-terminal mutations result in CEBPA variants that are unable to dimerize or bind DNA, thus rendering them inactive. Hence, CEBPA p30 homodimers are the sole entity with functional transcription factor activity in *CEBPA*[DM] AML. This is in contrast to normal hematopoietic cells where the full-length p42 isoform is predominantly expressed[2]. CEBPA p30 lacks two of three transactivation elements present in p42, but retains one transcriptional activating element and the basic-region leucine-zipper, which enables dimerization and DNA-binding[10]. CEBPA p30 has functions distinct from CEBPA p42 and can bind an isoform-specific set of enhancers and regulate the expression of downstream effector genes, such as *Nt5e* and *Msi2*[11,12]. Importantly, in the context of *CEBPA*[DM] AML, the *CEBPA*[NT] is hypermorphic, leading to higher levels of the transcription factor, and thus, increased binding to enhancers and subsequent deregulation of gene expression[11]. In line with these data, mice with CEBPA p30 expression driven from the endogenous *Cebpa* locus develop AML with full penetrance within a year[13].

Most patients with *CEBPA*[DM] AML also feature additional mutations in *GATA2*, *TET2*, *WT1*, *NRAS*, *FLT3*, or *CSF3R*[9]. Several of these mutations are found together with *CEBPA*[DM] more frequently than expected by the individual frequency of each mutation, while other combinations are statistically underrepresented. Recent studies have shed light on the molecular mechanisms underlying mutational cooperativity for some of the co-mutated genes, i.e. GATA2[14] and CSF3R[15], while mechanistic insight is still lacking for other subgroups of *CEBPA*[DM] AML. Of particular importance are mutations in the gene encoding the methylcytosine dioxygenase TET2 which, by converting 5-methylcytosine to 5-hydroxymethylcytosine, promotes DNA demethylation. *TET2* mutations (*TET2*[MUT]) are frequent in *CEBPA*[DM] AML cases and are associated with inferior prognosis[16,17]. Moreover, loss of *Tet2* has been implicated in accelerating and/or aggravating hematological malignancies in combination with several other recurrent gain-of-function and loss-of-function mutations[18–20], reflecting the importance of appropriately regulated DNA demethylation in normal hematopoiesis. Importantly, while *Tet2* loss alone only mildly affects hematopoiesis with myeloid skewing and increased competitiveness of HSCs[18], as well as the increased propensity of leukemic blasts to switch to a more stem-like phenotype[21], it does not induce overt leukemia per se[22–24]. Despite being extensively studied, mechanistic insights of how TET2 loss-of-function cooperates with other aberrations have been hampered by the fact that malignant cells have been compared to their normal, wild-type counterparts in many studies.

In the present work, we sought to overcome this limitation by comparing *CEBPA*-mutant AML in the presence and absence of additional mutations in *TET2*. By combining transcriptomic and epigenomic analyses of relevant in vitro and in vivo models as well as data from AML patients, we identified an intricate mechanism where TET2 loss-of-function rebalances *Gata2* expression levels in *Cebpa*[DM] AML, and hence drives an aggressive disease.

## Results

### *TET2* mutations impair outcome for patients with *CEBPA*-mutant AML

To validate previous reports on the spectrum of co-occurring mutations in *CEBPA*[DM] AML patients, we compiled data from 557 *CEBPA*[DM] cases and evaluated the co-occurrence of other known leukemia driver mutations[3–7,17]. *TET2* was the second most frequently co-mutated gene, with 1 in 5 *CEBPA*[DM] cases harboring *TET2* mutations (Fig. 1a; Supplemental Table 1). Importantly, the survival of *TET2*-mutant (*TET2*[MUT])

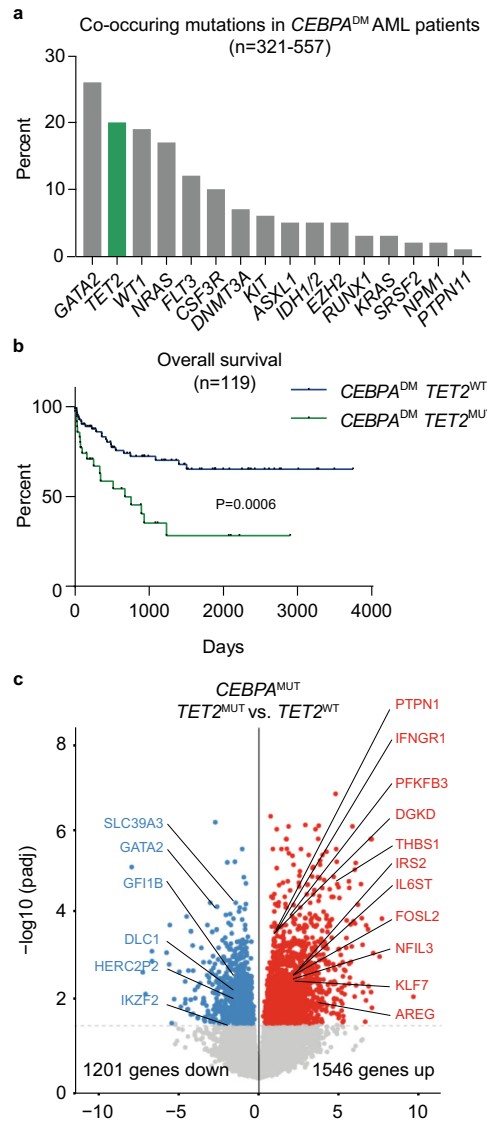

**Fig. 1 | *TET2* mutations impair outcome for patients with *CEBPA*-mutant AML. a** Frequency of co-occurring mutations in *CEBPA*[DM] AML cases, data aggregated from published cohorts[3–8,17] (321–557 cases; detailed in Supplemental Table 1). **b** Overall survival of *CEBPA*[DM] patients with wild-type (*TET2*[WT]; 84 patients) or mutated *TET2* (*TET2*[MUT]; 35 patients). The data were analyzed by Mantel-Cox Log-rank test. **c** Volcano plot depicting differentially expressed genes dependent on *TET2* mutational status in the cohort of *CEBPA*-mutant patients in the Beat AML dataset (*TET2*[WT] 11 and *TET2*[MUT] 5 patients). Differential analysis was performed with DESeq2 (*P* < 0.05). Source data are provided as a Source Data file.

*CEBPA*[DM] patients was significantly lower than *TET2* wild-type (*TET2*[WT]) *CEBPA*[DM] patients (Fig. 1b), consistent with previous reports[16], while the presence of *TET2* mutations did not cause a higher overall number of mutations in *CEBPA*[DM] patients (Supplemental Fig. 1a).

To investigate the functional consequences of *TET2* and *CEBPA* co-mutations, we analyzed RNA sequencing (RNA-seq) data from the Beat AML dataset[1]. We identified 1546 up- and 1201 downregulated genes in patients harboring a combination of *CEBPA* and *TET2* mutations when compared to *CEBPA*-mutant patients with wild-type *TET2* (Fig. 1c). Similarly, a slight overrepresentation of up-regulated genes was observed when comparing *CEBPA*[WT]*TET2*[MUT] patients to *CEBPA*[WT]*TET2*[WT] patients (601 up- and 527 downregulated). In line with the lower overall survival of *TET2*[MUT]*CEBPA*[DM] patients, pathways related to inflammation, hypoxia, and aggressive cancer were upregulated

in *CEBPA*-*TET2* co-mutated patients (Supplemental Fig. 1b). The over-representation of up-regulated genes associated with TET2 deficiency in *CEBPA*MUT (and *CEBPA*WT) patients is somewhat surprising, as increased DNA methylation upon TET2 loss would be expected to cause global transcriptional repression. However, other co-occurring mutations and residual DNA demethylase activity from the *TET2*WT allele may cause a more complex pattern of gene expression.

These findings indicate that mutations in *TET2* enhance the aggressiveness of *CEBPA*-mutant AML by deregulation of critical cellular pathways.

## TET2 deficiency accelerates *Cebpa*-mutant AML

To study the effect of *TET2* mutations in *CEBPA*DM AML in pathophysiologically relevant in vitro and in vivo models, we utilized cell and murine models in which expression of the p30 isoform is retained (*Cebpa*p30/p30 or *Cebpa*Δ/p30), while the normal p42 isoform of CEBPA is completely lost[13]. Since *TET2* is predominantly inactivated by loss-of-function mutations[25], we modeled *TET2* mutations either by the introduction of mutations with the CRISPR-Cas9 technology or by conditional knockout of the *Tet2* alleles.

First, we introduced *Tet2* mutations into a murine myeloid progenitor cell model (*Cebpa*p30/p30) (Fig. 2a). *Tet2*-targeted cells displayed a selective advantage, as they outcompeted *Cebpa*p30/p30 cells (Fig. 2b). Detailed analysis of the *Tet2* mutation that was associated with the proliferative advantage showed that the *Tet2* locus had acquired a +1 insertion in exon 3, which resulted in a downstream premature termination codon (Supplemental Fig. 2a, b). In line with this, clones isolated from the targeted cell pool exhibited strongly reduced TET2 protein expression (Supplemental Fig. 2c). Gene expression analysis revealed that *Tet2* loss in *Cebpa*p30/p30 cells caused downregulated expression of 916 genes, while only 540 genes were upregulated (Fig. 2c). Gene set enrichment analysis (GSEA) showed higher expression of MYC and E2F targets in *Cebpa*p30/p30 *Tet2*-mutated cells, consistent with their proliferative advantage (Supplemental Fig. 2d).

In summary, these data show that CRISPR/Cas9-induced TET2 loss provides a competitive advantage to myeloid progenitors expressing the oncogenic CEBPA variant p30.

Next, we wanted to assess the impact of hematopoietic expression of CEBPA p30 (*Cebpa*Δ/p30) with TET2-deficiency (*Tet2*−/−) on AML initiation in vivo. To do so, we transplanted lethally irradiated recipient mice with BM cells derived from mice with relevant allele combinations and, following hematopoietic reconstitution, induced hematopoietic-specific knockout of the *Cebpa* WT allele and/or the *Tet2* alleles (Fig. 2d). The combination of CEBPA p30 expression with *Tet2* loss led to an early expansion of myeloid (Mac1+) cells in the BM and blood compared to mice with hematopoietic cells featuring either alteration on its own (Fig. 2e; Supplemental Fig. 2e). Conforming to patient data and data obtained from *Cebpa*p30/30 cells, *Cebpa*Δ/p30*Tet2*Δ/Δ hematopoietic cells gave rise to AML with shorter latency than *Cebpa*Δ/p30*Tet2*+/+ cells, with a median survival of 23 and 43 weeks, respectively (Fig. 2f). Mice transplanted with *Cebpa*Δ/p30*Tet2*+/+ BM cells developed leukemia with similar latency as mice transplanted with *Cebpa*p30/p30 fetal liver cells[13]. This is consistent with the matching expression of *Cebpa* in these two contexts (1.1 ± 0.24 vs. 1.0 ± 0.13 (relative expression) in *Cebpa*p30/p30 and *Cebpa*Δ/p30*Tet2*+/+ AML blasts n = 3/group, respectively). TET2 deficiency alone (*Cebpa*Δ/+*Tet2*Δ/Δ) did not give rise to AML and cells which retained expression of the p42 isoform from one allele (*Cebpa*+/p30) only sporadically underwent leukemic transformation, in line with unaltered *Cebpa* expression levels in these cells (Fig. 2f; Supplemental Fig. 2f; 1.03 ± 0.14 vs. 1.0 ± 0.04 (relative expression) in *Cebpa*fl/p30 and *Cebpa*fl/+ cells n = 2–3/group, respectively). The transformed blasts expressed myeloid (Mac1+) and granulocytic (Gr1+) markers, confirming the myeloid origin of the leukemia (Supplemental Fig. 2g). The leukemias were transplantable into secondary recipients, and the shorter latency of the TET2-deficient *Cebpa*DM AML was

preserved in this setting (Supplemental Fig. 2h–i), indicating that TET2 not only has important tumor suppressive functions during malignant transformation but also during progression of AML.

We performed RNA-seq on *Cebpa*Δ/p30 (*Tet2* WT and knockout) AML blasts to assess changes in gene expression upon TET2 deficiency. Again, we found that the majority of differentially expressed genes was decreased in TET2-deficient AML blasts, with 176 down- vs. 58 upregulated genes (Fig. 2g). GSEA highlighted upregulation of genes involved in IL-6-JAK-STAT-signaling and hypoxia, in line with RNA-seq data from human *TET2*MUT*CEBPA*MUT cases (Supplemental Fig. 1b; Supplemental Fig. 2j). Furthermore, pathways related to cell cycle progression (G2M checkpoint and E2F targets) were enriched in TET2-deficient AML, indicating increased growth upon loss of TET2, consistent with the effects observed in the cell model (Supplemental Fig. 2d; Supplemental Fig. 2j). In line with this, we found that a higher frequency of *Cebpa*Δ/p30*Tet2*Δ/Δ blasts expressed the proliferation marker Ki67 (Fig. 2h). In addition, we also observed increased proliferative capacity of *Cebpa*Δ/p30*Tet2*Δ/Δ blasts compared to *Cebpa*Δ/p30*Tet2*+/+ blasts ex vivo. This difference was dependent on *Tet2* status, as the TET2 cofactor Vitamin C was able to mitigate proliferation of *Cebpa*Δ/p30*Tet2*+/+ but not of *Cebpa*Δ/p30*Tet*Δ/Δ cells (Supplemental Fig. 2k).

Collectively, these data show that TET2 deficiency accelerates the establishment and progression of CEBPA p30-driven AML in vivo.

## Loss of TET2 leads to reduced *Gata2* levels in *Cebpa*-mutant AML

To find conserved gene targets of the CEBPA-TET2 axis, we integrated the transcriptomic data from our in vitro and in vivo models with gene expression analyses from AML patients harboring *CEBPA* and *TET2* mutations. Three target genes exhibited downregulated expression in all three data sets; *FUT8*, *GATA2*, and *SIRT5* (Fig. 3a; Supplemental Fig. 3a–c).

Since the deregulation of these three genes was observed across species and differential experimental setups, we next aimed to investigate if their decreased gene expression was a direct result of TET2 deficiency. We therefore assessed chromatin accessibility and DNA methylation as a proxy for TET2 binding and activity[26]. Through assay for transposase-accessible chromatin sequencing (ATAC-seq), we identified 1809 differentially accessible regions in *Cebpa*p30/p30*Tet2*MUT vs. *Cebpa*p30/p30*Tet2*WT cells, and consistent with an activating effect of TET2, the majority of differential regions were less accessible in TET2-deficient cells (Fig. 3b). Half of the ATAC-seq peaks downregulated upon *Tet2* mutation were located in promoters, and these regions were enriched for GATA and NFAT motifs (Fig. 3c; Supplemental Fig. 3d). Using whole genome bisulfite sequencing (WGBS), we observed a global increase in DNA methylation in *Cebpa*Δ/p30*Tet2*Δ/Δ vs. *Cebpa*Δ/p30*Tet2*+/+ AML blasts, consistent with a loss of demethylase activity in *Tet2* knockout blasts (Fig. 3d). Increased DNA methylation was observed in promoter regions of genes whose expression were downregulated upon TET2 loss ( + 54%; Fig. 3e), while upregulated and not differently expressed genes did not show any marked changes. Strikingly, this pattern was not apparent when DNA methylation was evaluated across gene bodies (Supplemental Fig. 3e). Non-expressed genes exhibited equal increase in DNA methylation across promoters and gene bodies (Fig. 3e; Supplemental Fig. 3e). Since increased gene body methylation is not associated with gene repression[27], we evaluated whether a gain in gene body methylation was coupled to a gain in promoter methylation for the downregulated genes. In the presence of promoter hypermethylation, the bodies of down-regulated genes were more prevalently hypermethylated compared to neutral and up-regulated genes (34.8% [95%CI 18.8–55.11] vs. 18.1% [16.4–19.9], p = 0.0427). While, in the absence of promoter hypermethylation, the bodies of up-regulated genes tended to be hypermethylated compared to neutral and down-regulated genes (13.8% [95%CI 7.2–24.9] vs. 7.1% [6.8–7.4], p = 0.0518). Thus, loss of TET2 in *Cebpa*DM cells caused decreased chromatin accessibility and increased methylation of DNA in promoters of TET2-responsive genes,

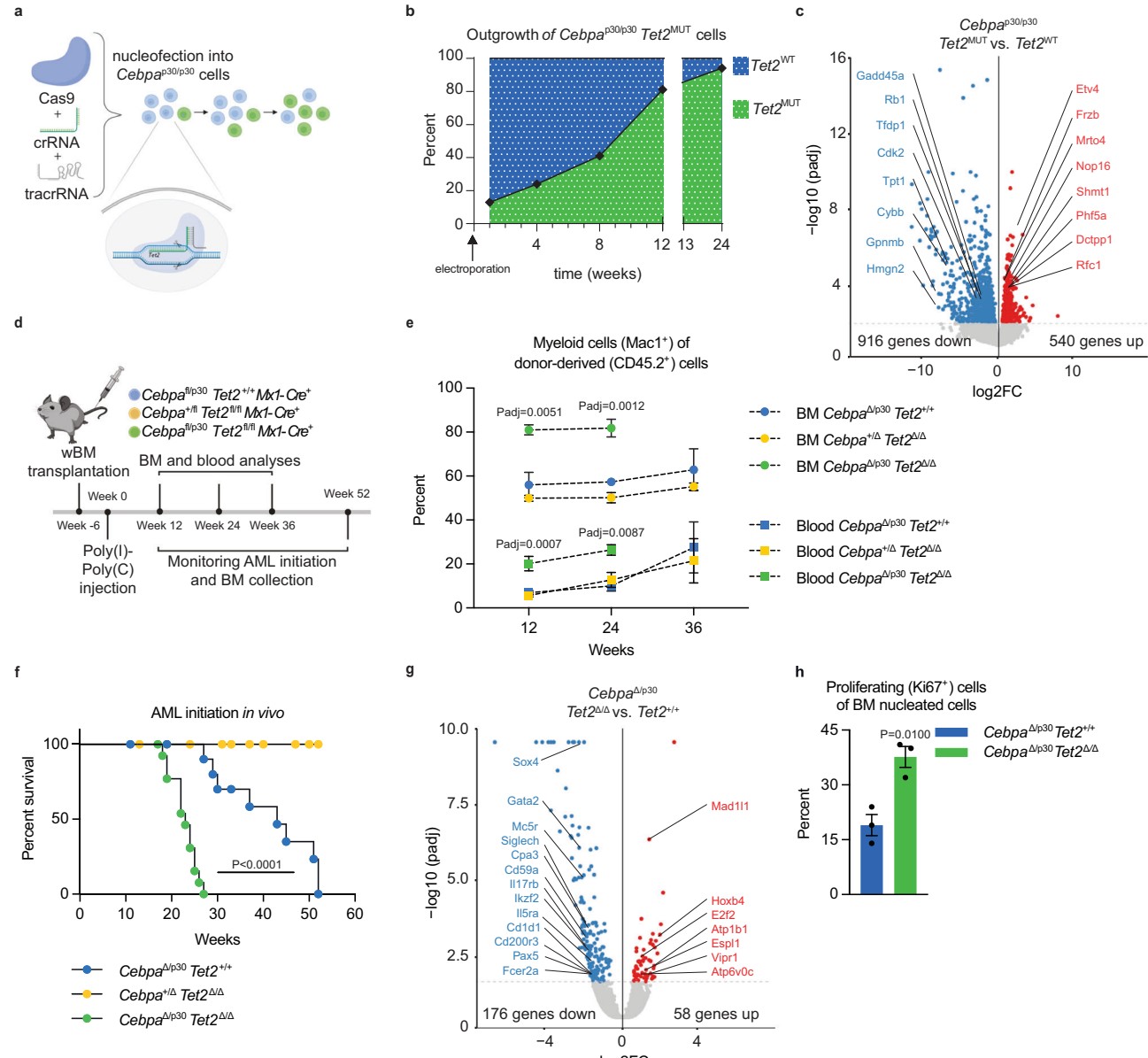

**Fig. 2 | TET2 deficiency accelerates *Cebpa*-mutant AML. a** Schematic representation of generation of *Tet2*-knockout clones with CRISPR/Cas9. The illustration was created with BioRender.com. **b** Proliferative outgrowth of *Cebpa*^p30/p30 cells with *Tet2* indels. **c** Volcano plot depicting differentially expressed genes dependent on the *Tet2* mutational status in *Cebpa*^p30/p30 cells (*Tet2*^WT 7 and *Tet2*^MUT 5 clones). Differential analysis was performed with DESeq2 (*P* < 0.05). **d** Experimental setup for evaluating the effect of *Tet2*-deficiency (*Tet2*^Δ/Δ) in *Cebpa*^DM AML initiation in vivo. The illustration was created with BioRender.com. **e** Myeloid (Mac1^+) contribution of donor-derived blood and bone marrow (BM) cells evaluated after BM transplantation and Cre-LoxP recombination. (Blood samples: Week 12; 6 mice per group. Week 24; *Cebpa*^Δ/p30*Tet2*^+/+ and *Cebpa*^+/Δ*Tet2*^Δ/Δ 6 mice per group and *Cebpa*^+/Δ*Tet2*^Δ/Δ 3 mice. Week 36; 3 mice per group. BM

samples: 3 mice per group.) Data are presented as mean±SEM and analyzed by one-way-ANOVA followed by Dunnett's multiple comparisons correction. **f** Survival of lethally irradiated recipient mice after BM transplantation and Cre-LoxP recombination (*Cebpa*^Δ/p30*Tet2*^+/+ 12 mice, *Cebpa*^Δ/p30*Tet2*^Δ/Δ 14 mice, and *Cebpa*^+/-*Tet2*^Δ/Δ 14 mice). The data were analyzed by Mantel-Cox Log-rank test. **g** Volcano plot depicting differentially expressed genes dependent on *Tet2* deficiency status in *Cebpa*^Δ/p30 leukemic blasts (samples from 3 mice per group). Differential analysis was performed with DESeq2 (*P* < 0.05). **h** Frequency of proliferating (Ki67^+) cells in BM of moribund recipient mice (specimens from 3 mice per group). Data are presented as mean±SEM and analyzed by a two-tailed unpaired *t*-test. Source data are provided as a Source Data file.

consistent with previous reports showing that TET2 binding is enriched in promoters of TET2-regulated genes[28].

To identify direct CEBPA-TET2 gene target(s), we evaluated the previously identified conserved candidates based on changes in DNA methylation of their promoters. Out of the three target genes, only the gene encoding the transcription factor GATA-binding factor 2 (GATA2) showed a gain of DNA methylation in the promoter of the gene variant 2 (*Gata2 V2*) upon TET2 deficiency (+46%; Fig. 3f). In line with this,

specifically the *Gata2 V2* mRNA isoform was downregulated in TET2-deficient *Cebpa*^DM AML blasts (−86%; Fig. 3g), while changes in mRNA expression and promoter methylation of *Gata2 V1* did not reach statistical significance (Fig. 3f, g).

In summary, these analyses identify *Gata2* (locus overview in Fig. 3h) as a conserved target of the CEBPA-TET2 axis across several settings. TET2 deficiency causes increased DNA methylation of the *Gata2* promoter, resulting in reduced mRNA expression.

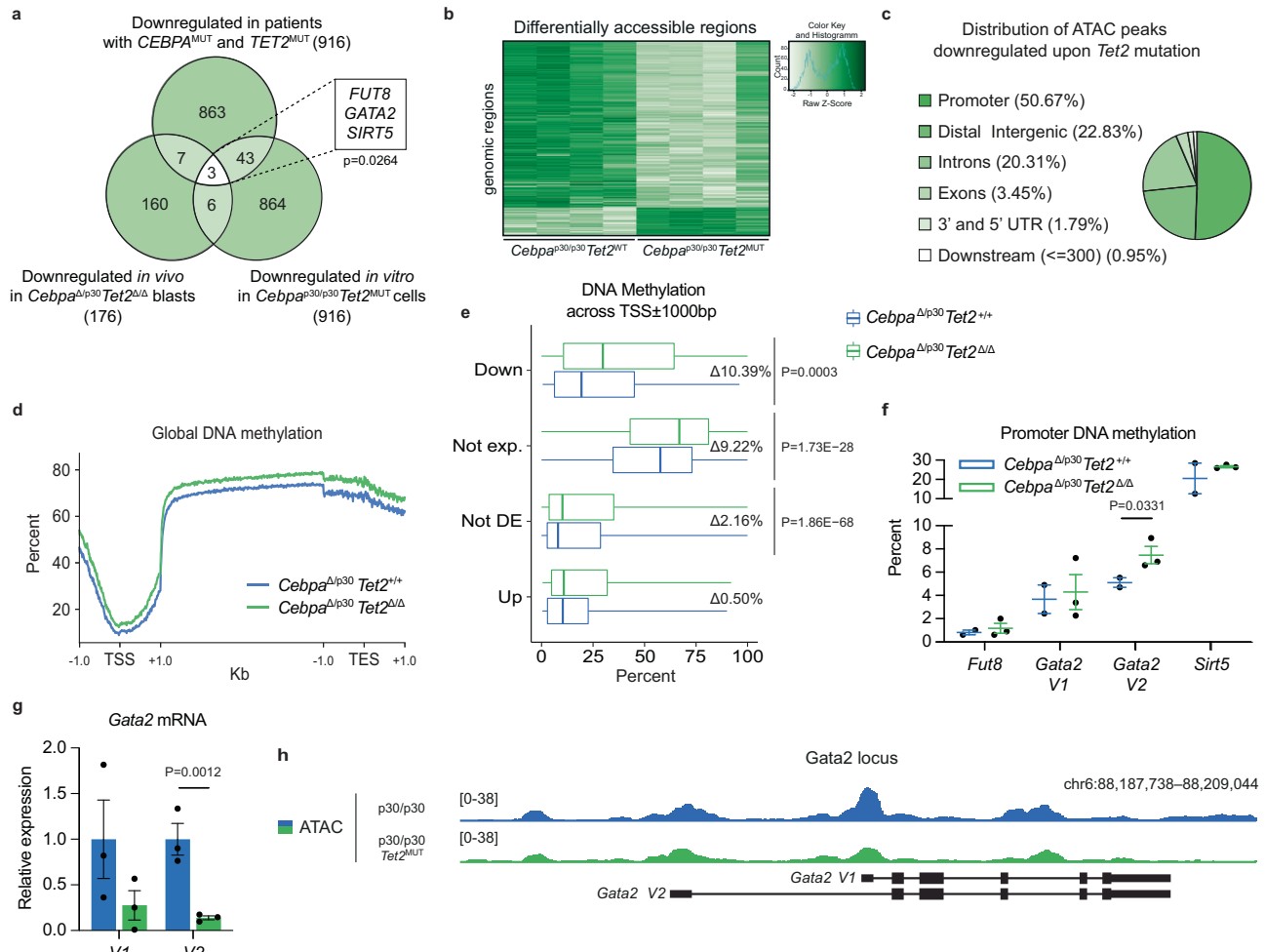

**Fig. 3 | Loss of TET2 leads to reduced *Gata2* levels in *Cebpa*-mutant AML.**
**a** Conserved targets of the CEBPA-TET2 axis visualized in a Venn-diagram of downregulated genes in *CEBPA-TET2* co-mutated AML overlaid with corresponding data from *Tet2*-deficient in vivo and in vitro models of *Cebpa*^DM AML (*P* = 0.0264 vs. number of overlapping genes expected by random distribution assessed by Wilson/Brown binominal test). **b** Heatmap of differentially accessible regions assessed by assay for transposase-accessible chromatin sequencing (ATAC-seq; FDR < 0.05), and **c** genomic distribution of downregulated peaks (FDR < 0.05, Log2FC < 0) upon *Tet2* mutation (4 clones per group). Differential analysis was performed with Diff-Bind and region enrichment analysis with GREAT. **d** Representative genome wide DNA-methylation status in leukemic blasts from the in vivo model assessed by whole genome bisulfite sequencing (WGBS) showing frequency of methyl-cytosine (mC) across the transcription start site (TSS) ±1000 base pairs, gene body scaled to 4000 base pairs, and transcription termination site (TES) ±1000 base pairs. Methylation analysis was performed with Bismark and visualized using deepTools.

**e** Median and interquartile range of percent mC at promoters of down- (*n* = 172), not expressed (*n* = 6539), not differentially expressed (not DE; n = 14759) and up-regulated (*n* = 57) genes (averaged data generated from 2 *Cebpa*^Δ/p30*Tet2*^+/+ and 3 *Cebpa*^Δ/p30*Tet2*^Δ/Δ mice). Whiskers indicates max–min and data were analyzed by two-tailed unpaired *t*-test. **f** Promoter DNA methylation of conserved target genes in leukemic blast (samples from 2 *Cebpa*^Δ/p30*Tet2*^+/+ and 3 *Cebpa*^Δ/p30*Tet2*^Δ/Δ mice). Data are presented as mean±SEM. The data were log-transformed and analyzed by two-tailed unpaired *t*-test. **g** *Gata2* variant mRNA expression in *Cebpa*^Δ/p30*Tet2*^Δ/Δ and *Cebpa*^Δ/p30*Tet2*^+/+ leukemic blasts (samples from 3 mice per group). Data are presented as mean ± SEM. The data were log-transformed and analyzed by a two-tailed unpaired *t*-test. **h** Schematic genomic view of the *Gata2* locus, including representative examples of assay for transposase-accessible chromatin using sequencing (ATAC-seq) in *Cebpa*^p30/p30 cells. Source data are provided as a Source Data file.

## Moderate *Gata2* reduction increases competitiveness of *Cebpa*-mutant AML

GATA2 is an essential transcription factor for hematopoietic cells and has profound effects on HSC maintenance. Moreover, it is recurrently mutated in AML[29,30] and *GATA2* lesions are overrepresented in *CEBPA*^DM AML[8,16,31–33]. Given these critical roles of GATA2, we next examined the consequences of reduced GATA2 levels in *CEBPA*^DM AML.

To test if reduced *Gata2* expression would provide a competitive advantage in vivo, we set up an RNA-interference (RNAi) based competition assay (Fig. 4a) utilizing established *Cebpa*^p30/p30 leukemia cells, in which both *Cebpa* (+56–73%) and *Gata2* (+45–56%) levels are increased modestly compared to primary *Cebpa*^Δ/p30*Tet2*^+/+ blasts[11,34]. First, we identified four short hairpin RNAs (shRNA) which lowered *Gata2* expression to a varying degree (Fig. 4b). Upon transplantation of

shRNA-expressing cells, we observed a non-monotonic relationship between *Gata2* expression levels and competitiveness, as measured by sh*Gata2*-to-shControl ratios. While efficient downregulation of *Gata2* expression did not provide any competitive advantage to *Cebpa*^DM cells, moderate silencing imposed a three-fold increase in their ability to compete (Fig. 4c, d). Repetition of this experiment including only the most and least efficient shRNAs in a separate experiment yielded similar results (Supplemental Fig. 4a, b). These results were mirrored by increased expression of the proliferation marker *Ki67* in cells expressing the least efficient *Gata2*-targeting shRNA but not the most efficient one (Supplemental Fig. 4c). To test if the same effects are observed in an in vitro setting, we targeted *Gata2* in *Cebpa*^p30/p30 cells using the CRISPR/Cas9 approach. *Gata2*-targeted cells showed a proliferative advantage over *Gata2*^WT cells, leading to their outgrowth

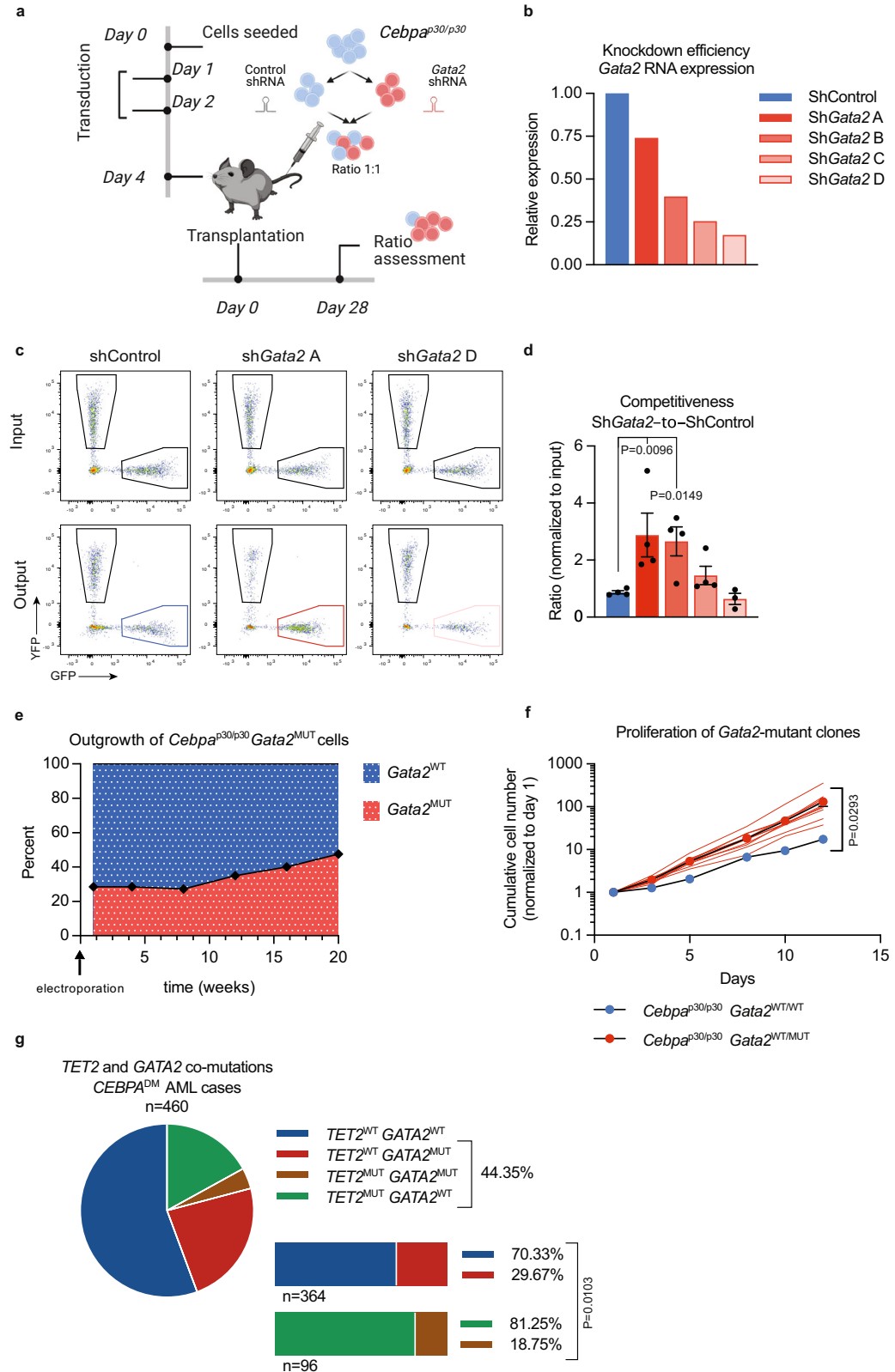

(Fig. 4e, f). In accordance with previously published data that complete loss of *Gata2* expression results in a loss of competitiveness[35–37], we found that only clones with heterozygous *Gata2* inactivation were viable, while clones with homozygous mutations in *Gata2* could not be recovered (Supplemental Fig. 4d).

If the pro-leukemogenic effect of *TET2* mutations was, at least partly, caused by lowering *GATA2* expression, we reasoned that concomitant mutations in both genes would be redundant and thus, the pattern of *TET2* and *GATA2* mutations would be mutually exclusive. Indeed, *TET2*^MUT*CEBPA*^DM AML cases showed a lower frequency of *GATA2* mutations than expected from the frequency of *GATA2* mutations in *TET2*^WT*CEBPA*^DM AML cases (Fig. 4g; Supplemental Table 2a), which was also true for all AML cases (3.5% in TET2^MUT vs. 8.9% in TET2^WT; Supplemental Fig. 4e; Supplemental Table 2b). Importantly,

**Fig. 4 | Moderate *Gata2* reduction increases competitiveness of *Cebpa*-mutant AML. a** Experimental setup for evaluating the effect of *Gata2* knockdown, via short hairpin RNA (shRNA) mediated silencing, on *Cebpa*[p30/p30] leukemic cells in a competitive in vivo assay. The illustration was created with BioRender.com. **b** *Gata2* mRNA in *Cebpa*[p30/p30] leukemic cells prior to transplantation. **c** Representative flow cytometry profiles of input and output of shControl (no knockdown), sh*Gata2*A (low knockdown), and sh*Gata2*D (high knockdown). **d** Competitive advantage of targeting shRNA (GFP[+]) vs. non-targeting shRNA (YFP[+]) cells in vivo assessed as by flow cytometry (Control 4, sh*Gata2*A 4, sh*Gata2*B 4, sh*Gata2*C 4, and sh*Gata2*D 3 mice). Data are presented as mean±SEM. Data were log-transformed and analyzed by one-way-ANOVA followed by Dunnett's multiple comparisons correction.

**e** Experimental setup for *Gata2* CRISPR/Cas9 mutagenesis in *Cebpa*[p30/p30] cells, and outgrowth of heterozygous mutated clones. Percentages of *Gata2* mutated clones are indicated. **f** Growth curve of *Cebpa*[p30/p30] clones with *Gata2* mutation (*Cebpa*[p30/p30]*Gata2*[+/MUT], n = 10) or wild type *Gata2* (*Cebpa*[p30/p30]*Gata2*[+/+], n = 3). Data are presented as mean±SEM and analyzed by two-tailed unpaired *t*-test. Red lines mark individual mutated clones. **g** Presence or absence of *GATA2* mutations (*GATA2*[MUT]) in *CEBPA* double mutated (*CEBPA*[DM]) AML cases with or without *TET2* mutations (*TET2*[MUT]) in aggregated data from published cohorts[3–5,7,8,17] (detailed in Supplemental Table 2a). Data were analyzed by Wilson/Brown binominal test. Source data are provided as a Source Data file.

whereas mutations in *WT1* followed the same pattern as *TET2*, *CSF3R* mutations appeared in equal frequency between *TET2*[MUT]*CEBPA*[DM] and *TET2*[WT]*CEBPA*[DM] AML cases, and *ASXL1* mutations were increased in *TET2*[MUT]*CEBPA*[DM] AML (Supplemental Fig. 4f; Supplemental Table 2c–e). While we favor a functional redundancy model, we cannot exclude that co-mutation of *TET2* and *GATA2* could induce synthetic lethality in AML cells, as *Gata2*-loss has been shown to induce terminal myeloid differentiation[37].

Altogether, our data suggest that loss of TET2 in *Cebpa*[DM] AML causes a moderate decrease in *Gata2* expression, which in turn increases the competitive fitness of the leukemia. Hence, this indicates that *TET2* and *GATA2* mutations are partially redundant in *CEBPA*[DM] AML, providing a mechanistic rationale for the mutational spectrum observed in this AML entity.

### Increased CEBPA p30 binding to the *Gata2* distal hematopoietic enhancer drives expression of *Gata2* via TET2

We next asked if *GATA2* expression is dependent on *CEBPA* mutational status. To this end, we exploited published transcriptomics data from human and mouse *CEBPA*[DM] AML[11]. *GATA2* expression was increased in human *CEBPA*[DM] leukemic granulocyte/monocyte progenitors (GMPs) compared to GMPs from healthy donors (+77%; Supplemental Fig. 5a). Correspondingly, *Gata2* was upregulated in murine *Cebpa*[p30/p30] leukemic GMPs as compared to normal GMPs (+43%; Fig. 5a). Likewise, analysis of AML patient data from the BEAT AML study[1], revealed that both *CEBPA* and *GATA2* expression were increased in *CEBPA*[NT] AML compared to *CEBPA*[WT] AML (+91% and +37%, respectively), while *GATA2* expression was reverted to *CEBPA*[WT] level in *CEBPA*[NT]*TET2*[MUT] AML (Supplemental Fig. 5b, c). Since CEBPA is known to exert its transcription factor activity by binding to enhancers and thereby promote gene expression[38], we assessed the binding of CEBPA to the crucial *Gata2* distal hematopoietic enhancer (*G2*DHE; −77 kb in mouse) that governs *Gata2* expression in hematopoietic stem and progenitor cells including GMPs[11,39]. Notably, we found substantially increased levels of CEBPA bound to the *G2*DHE in *Cebpa*[p30/p30] leukemic GMPs compared to their normal counterparts (+147%; Fig. 5b), while the binding levels associated with other known proximal and distal *cis*-regulatory elements of the *Gata2* gene were unchanged (Supplemental Fig. 5d, e). Additionally, TET2 showed significant binding to the *G2*DHE in *Cebpa*[p30/p30] cells (Fig. 5c). However, DNA methylation at the *G2*DHE was low and unaltered upon *Tet2* loss (Supplemental Fig. 5f). Importantly, CEBPA binding, as assessed by ChIP-qPCR, was not altered by introduction of *Tet2* mutations in *Cebpa*[p30/p30] cells (Supplemental Fig. 5g).

These results prompted us to test if CEBPA binding to the *G2*DHE modulates *Gata2* expression in *Cebpa*[DM] AML. We deleted 250–500 bp fragments of the *Gata2* enhancer encompassing the CEBPA binding site using the CRISPR/Cas9 approach in *Cebpa*[p30/p30] cells in vitro. Expression of total *Gata2* mRNA, as well as both individual transcript variants, was decreased upon targeting the genomic region with strong CEBPA binding compared

to non-targeting control (Fig. 5c–e, Supplemental Fig. 5h–k). In contrast, *Gata2* expression was unchanged when *G2*DHE deletions were introduced in *Cebpa*[p30/p30]*Tet2*[MUT] cells (Supplemental Fig. 5l, m). Combined, these data suggest that CEBPA binding to the *G2*DHE is important for promoting *Gata2* expression in *Cebpa*[DM] AML. Further, the *G2*DHE has been shown to primarily regulate expression of the hematopoietic specific *Gata2 variant 2* (*V2*)[40,41], conforming with our data that particularly the *Gata2 V2* promoter displayed an increase in DNA methylation and that the *Gata2 V2* mRNA was downregulated in TET2-deficient *Cebpa*[DM] AML blasts (Fig. 3f–g).

Next, we tested if the reduction of CEBPA in AML cells influenced the expression and promoter DNA methylation of *Gata2 V2*. Given the dependence of *CEBPA*[DM] AML on CEBPA for survival and maintenance, we utilized MLL-fusion-driven AML, in which CEBPA is dispensable for the maintenance of established leukemia[42]. Cre-mediated loss of *Cebpa* in leukemic cells expressing the inducible MLL-AF9 fusion-protein (*iMLL-AF9*[+]*Cebpa*[Δ/Δ]; Fig. 5f) caused reduced *Gata2 V2* mRNA levels compared to control cells (*iMLL-AF9*[+]*Cebpa*[fl/fl]; (−72%; Fig. 5g). Importantly, the methylation frequency of the CpG island located at the *Gata2 V2* promoter was increased in two separate leukemic lines (+186%; Fig. 5h; Supplemental Fig. 5n), suggesting that *Gata2 V2* mRNA expression is regulated via the CEBPA-TET2 axis. Finally, we assessed TET2 binding to the *G2*DHE upon *Cebpa* knockdown in *Cebpa*[p30/p30] cells using ChIP-qPCR (Fig. 5i). Notably, we observed decreased TET2 binding to the *G2*DHE in cells expressing sh*Cebpa* compared to cells expressing control shRNA, verifying that CEBPA is important for recruitment of TET2 to the *G2*DHE (Fig. 5j).

In light of these findings, we asked whether elevated CEBPA level and not the *CEBPA* mutation(s) per se, drives the selective pressure for *GATA2* and/or *TET2* loss in AML to achieve moderate *GATA2* levels that are optimal for leukemia growth. We therefore stratified AML cases in the Beat AML cohort[1] based on *CEBPA* expression and assessed their *GATA2* and *TET2* mutational status. Indeed, the frequency of *GATA2* and/or *TET2* mutations was three-fold higher in *CEBPA*[HIGH] AML compared to the *CEBPA*[LOW] samples (Fig. 5k). In line with previous data showing a hypermorphic effect of *CEBPA*[DM][11], the *CEBPA*[HIGH] group contained the majority of the *CEBPA*-mutant cases in the cohort (82 and 100% of *CEBPA*[SM] and *CEBPA*[DM], respectively), while none of the cases in the *CEBPA*[LOW] group were *CEBPA*-mutated.

In conclusion, our data show that elevated CEBPA binding to the *G2*DHE, driven by the hypermorphic effect of *Cebpa*[NT], increases TET2-mediated demethylation of the *Gata2* promoter, which leads to elevated *Gata2* levels in *Cebpa*[DM] AML. In this context, *Cebpa*[DM] AML cells gain a competitive advantage by loss of TET2, which in turn promotes an increase in DNA methylation at the *Gata2* promoter resulting in the rebalancing of *Gata2* levels.

### Demethylating treatment restores *Gata2* expression and prolongs survival in TET2-deficient *Cebpa*-mutant AML

Finally, we investigated if treatment with the demethylating agent 5-azacytidine (5-AZA) would be beneficial in TET2-deficient *CEBPA*[DM] AML. Ex vivo treatment with 5-AZA restored *Gata2* expression in

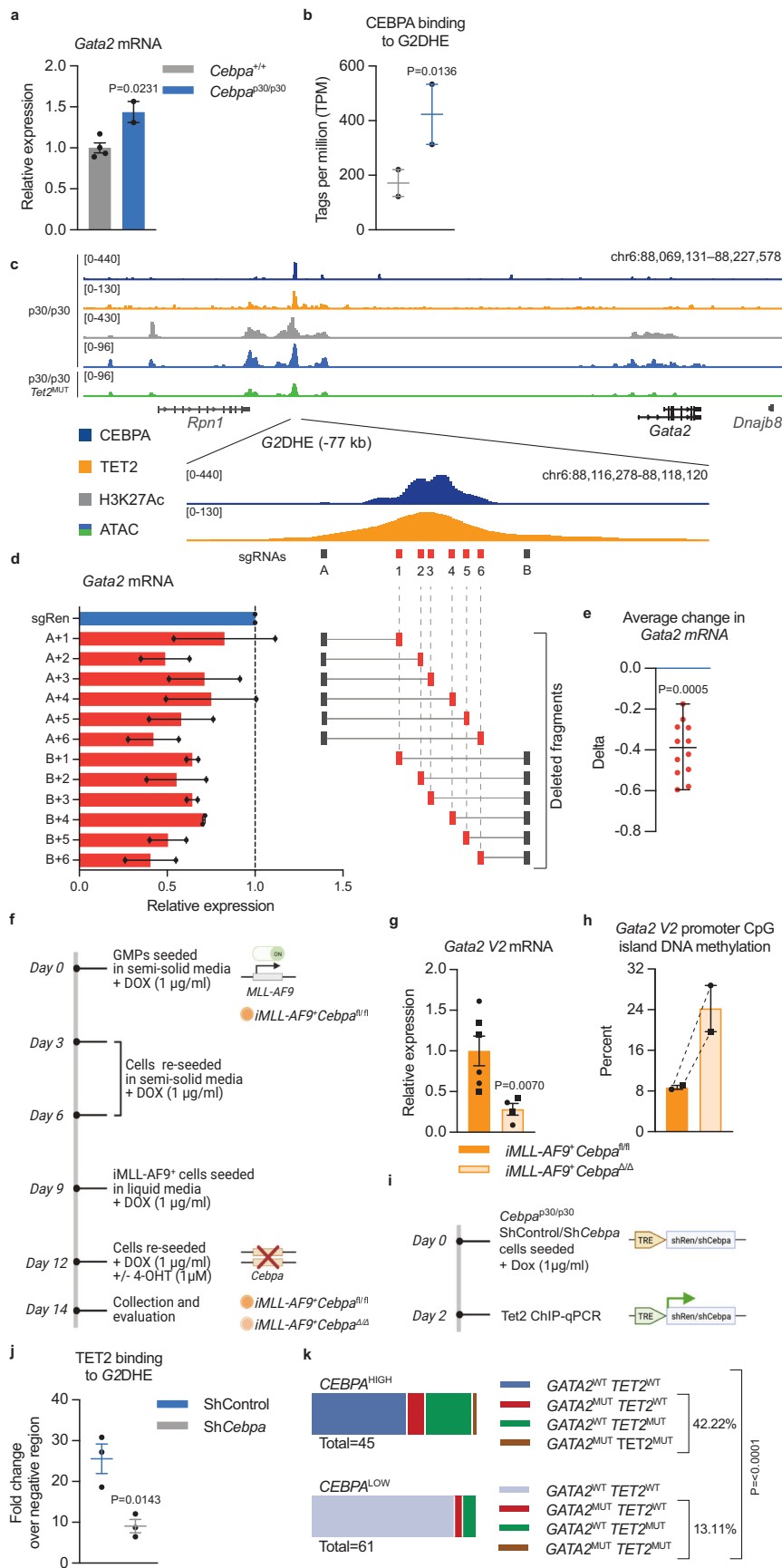

**Fig. 5 | Increased CEBPA p30 binding to the *Gata2* distal hematopoietic enhancer drives expression of *Gata2* via TET2. a** *Gata2* mRNA expression in mouse *Cebpa*[p30/p30] leukemic granulocyte/monocyte progenitors (GMPs) vs normal GMPs (samples from 4 *Cebpa*[+/+] and 2 *Cebpa*[p30/p30] mice) and, **b** CEBPA binding to the *Gata2* distal hematopoietic enhancer (*G2*DHE; −77 kb) region (samples from 2 mice per group), data from Jakobsen et al.[11]. Data are presented as mean ± SEM. Differential analysis was performed with DESeq2 (*P* < 0.05). **c** Schematic genomic view of the *Gata2* distal hematopoietic enhancer (*G2*DHE), including normalized chromatin immunoprecipitation sequencing (ChIP-seq) signal of CEBPA (data from Heyes et al.[12]), TET2 and H3K27Ac (data from Heyes et al.[12]), as well as assay for transposase-accessible chromatin using sequencing (ATAC-seq) in *Cebpa*[p30/p30] cells without (light blue) and with (green) mutation in *Tet2*. **d** *Gata2* mRNA levels in response to targeting of the *G2*DHE by CRISPR-Cas9 in *Cebpa*[p30/p30] cells in vitro using indicated sgRNAs and **e** the averaged change in *Gata2* mRNA levels of the 12 deletions (averaged data from 2 separate experiments). Data are presented as median ± range and analyzed by two-tailed Wilcoxon signed-rank test. **f** Experimental setup for evaluating the effects of *Cebpa* knockout on *Gata2 V2* mRNA expression and DNA methylation of the CpG island at the promoter of *Gata2*

*V2* in MLL-fusion driven AML (*iMLL-AF9*). The illustration was created with BioRender.com. **g** *Gata2 V2* mRNA expression (leukemic cell lines generated from 2 separate mice were assayed on 2 separate days in 2–3 technical replicates each). Data are presented as mean ± SEM and the individual cell lines are indicated by circles or squares. Data were log-transformed and analyzed by two-tailed unpaired *t*-test. **h** DNA methylation of the *Gata2* V2 promoter CpG-island (2 separate leukemic cell lines). Data are presented as median ± range and the individual cell lines are indicated by circles or squares. **i** Experimental setup for evaluating the effects of *Cebpa* knockdown on TET2 binding to the *G2*DHE in *Cebpa*[p30/p30] cells with inducible expression of shRNA targeting *Cebpa* and control (*Renilla*), respectively. The illustration was created with BioRender.com. **j** TET2 binding to the *G2*DHE assessed by ChIP-qPCR (3 replicates per condition). Data are presented as mean ± SEM and analyzed by two-tailed unpaired *t*-test. **k** Frequency of *GATA2* and/or *TET2* mutations (*GATA2*[MUT] and TET2[MUT], respectively) in *CEBPA* high expressing (*CEBPA*[HIGH]; 45 cases) vs. *CEBPA* low expressing (CEBPA[LOW]; 61 cases) AML cases, data from Beat AML cohort[1]. The distributions of *GATA2*[WT]*TET2*[WT] vs. *GATA2*[MUT] and/or *TET2*[MUT] cases were analyzed by Wilson/Brown binominal test. Source data are provided as a Source Data file.

*Cebpa*[Δ/p30]*Tet2*[Δ/Δ] blasts to levels observed in *Cebpa*[Δ/p30]*Tet2*[+/+], while 5-AZA treatment did not affect *Gata2* levels in *Cebpa*[Δ/p30]*Tet2*[+/+] cells (Supplemental Fig. 6a). Moreover, 5-AZA decreased the viability of blasts from both genotypes, although to a higher degree in the TET2-deficient setting (−82% and −40%, respectively, *p* < 0.01; Supplemental Fig. 6b).

To evaluate if the enhanced effect of 5-AZA treatment in TET2-deficient AML would also hold true in vivo, mice were transplanted with *Cebpa*[Δ/p30]*Tet2*[Δ/Δ] or *Cebpa*[Δ/p30]*Tet2*[+/+] AML blasts and treated with 5-AZA for three consecutive days after disease establishment (Fig. 6a). While the blast frequency of TET2-deficient *Cebpa*[Δ/p30] AML decreased upon 5-AZA treatment (− 62%; Fig. 6b), the treatment did not significantly decrease the frequency of TET2-proficient cells. Furthermore, 5-AZA treatment restored *Gata2* levels in *Cebpa*[Δ/p30]*Tet2*[Δ/Δ] blasts in vivo to the same level as in *Cebpa*[Δ/p30]*Tet2*[+/+] blasts (Fig. 6c). Intriguingly, two out of three individual *Cebpa*[Δ/p30]*Tet2*[Δ/Δ] leukemic clones (A + B) responded to 5-AZA treatment with a pronounced increase of *Gata2* levels and concomitant reduction of myeloid blasts, while one clone (C) appeared partially refractory to 5-AZA treatment, with limited increase of *Gata2* and no reduction of leukemic burden (Fig. 6b, c). Importantly, a longer intermittent 5-AZA treatment prolonged the survival of mice transplanted with *Cebpa*[Δ/p30]*Tet2*[Δ/Δ] blasts from one of the responding clones (A) (median survival +22%; Fig. 6d, e), while it did not affect disease latency of mice transplanted with *Cebpa*[Δ/p30]*Tet2*[+/+] blasts (A).

In summary, we show that the demethylating agent 5-AZA can restore *Gata2* expression levels in TET2-deficient *Cebpa*[DM] AML to that of TET2-proficient *Cebpa*[DM] AML, and concomitantly reduce leukemic burden and prolong survival of mice transplanted with TET2-deficient *Cebpa*[DM] leukemic blasts.

## Discussion

Mutational cooperativity is a fundamental driver of cancer development, progression, and aggressiveness. For *CEBPA*[DM] AML, co-occurring lesions have been found in genes such as *GATA2*, *TET2*, *WT1*, *FLT3*, and *CSFR3*. While the mechanistic basis for the cooperation between *CEBPA* and *GATA2/CSFR3* mutations has been investigated using mouse models[14,15], we have very little insight into why other lesions, such as those in *TET2*, are overrepresented in *CEBPA*[DM] AML. Here, we show that TET2 loss-of-function in *CEBPA*[DM] AML leads to an aggressive disease phenotype by rebalancing the increased and sub-optimal levels of *GATA2* that are induced by hypermorphic *CEBPA*[NT] mutations driving CEBPA-p30 isoform expression (see model in Fig. 7a). Specifically, loss of TET2 binding to the hematopoietic-specific *G2*DHE enhancer results in increased DNA methylation in the promoter region of the hematopoietic-specific *Gata2* isoform (*Gata2 V2*). This

proleukemic effect of TET2 loss can be reversed by the demethylating agent 5-AZA, suggesting that this could be a potential treatment option in *CEBPA*[DM]*TET2*[MUT] patients. Altogether, our work proposes that CEBPA-mutant AMLs acquire additional lesions in genes such as *GATA2* and *TET2* to reestablish balanced *GATA2* levels that permit leukemia development and progression.

Our work highlights the central importance of GATA2 regulation in *CEBPA*-mutant AML. Specifically, we show that *GATA2* is a conserved target gene of CEBPA and TET2. Furthermore, the elevated levels of the CEBPA p30 variant likely mediate *GATA2* upregulation in CEBPA-mutant AML. The increased expression of *Gata2* is counteracted by loss of *TET2* in vitro and in vivo models of *Cebpa*[DM] AML as well as in *CEBPA*·*TET2* co-mutated patients. This is accompanied by the gain of *Gata2* promoter DNA methylation. These findings are consistent with previous data showing that *Gata2* expression is TET2-dependent, as *Gata2* was downregulated in various *Tet2* knockout settings and that forced expression of *Gata2* decreased the competitiveness of both normal and malignant TET2-deficient cells[28,43–45]. Further paralleling our data, TET2 deficiency in the context of *Flt3*[ITD] AML has been shown to accelerate leukemia by hypermethylation and consequent silencing of the *Gata2* locus[43].

Strikingly, we found that while moderate reduction of *Gata2* expression increased competitiveness in *Cebpa*[DM] AML both in vivo and in vitro, leukemia cells remain critically dependent on residual GATA2 function. Indeed, homozygous *Gata2* lesions induced a strong inhibitory effect on *Cebpa*[DM] AML in vitro[37], which was also observed in other AML subtypes as well as in normal hematopoietic stem cells[36,46–48]. These findings are corroborated by a substantial body of genetic evidence supporting the importance of GATA2 regulation in *CEBPA*-mutant AML. First, heterozygous *GATA2* lesions frequently co-occur with *CEBPA*[DM][4,8,16,17,31–33,49–52]. Secondly, *GATA2* allele-specific expression is strongly associated with *CEBPA*[DM] AML and is neither found in AML with reduced *CEBPA* expression (i.e. t(8;21)) nor in *CEBPA*-silenced AML[53]. Thirdly, *TET2*[MUT] and *GATA2*[MUT] rarely co-occur in *CEBPA*[DM] AML. Finally, we showed that mutations in *GATA2* and *TET2* are overrepresented in AML cases with high *CEBPA* expression. This supports the notion that unfavorable, high *GATA2* levels in AML promoted by the CEBPA-TET2 axis are not limited to *CEBPA*[DM] AML, but also include cases where *CEBPA* expression is high for other reasons. Further, this model also suggests that a major proleukemic effect of TET2 deficiency is to rebalance *GATA2* levels in the context of *CEBPA*[DM] AML (see Fig. 7b).

*GATA2* expression is mainly driven by the conserved *G2*DHE in normal myeloid progenitors and leukemic blasts by promoting expression from the hematopoietic specific *Gata2 V2* promoter[39,40,44,54]. Our data demonstrate that CEBPA plays a key role in

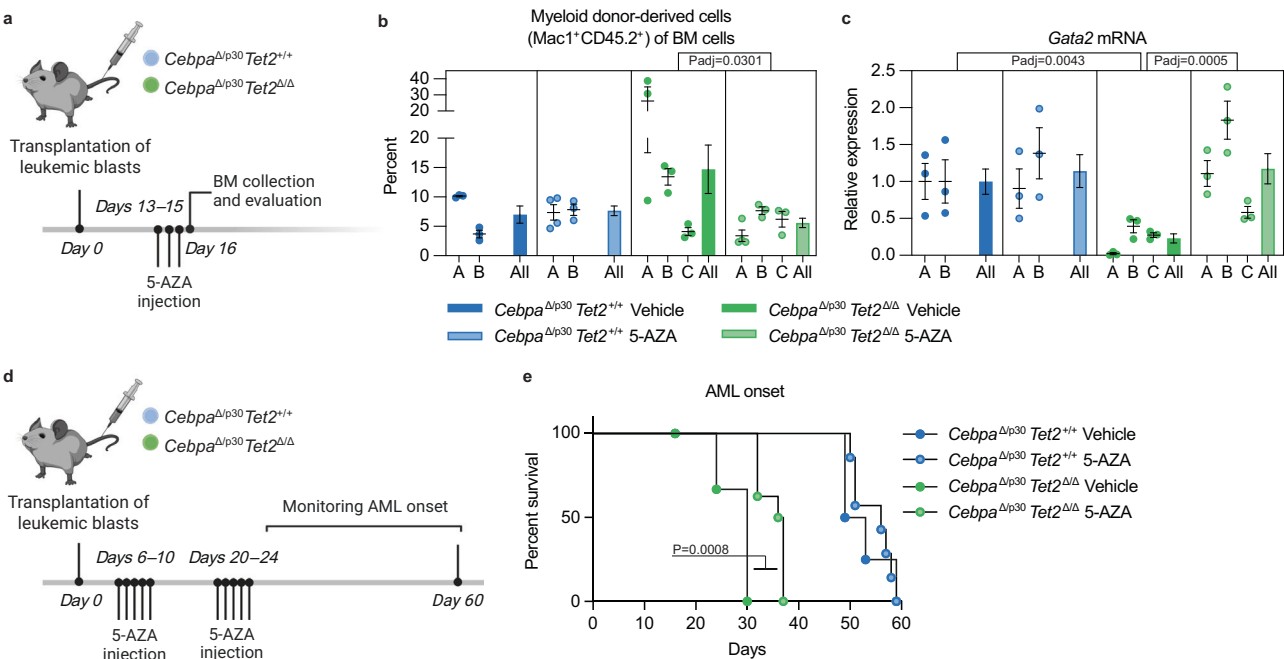

**Fig. 6 | Demethylating treatment restores *Gata2* expression and prolongs survival in TET2-deficient *Cebpa*-mutant AML. a** Experimental setup for evaluating the effect of short-term 5-azacytidine (5-AZA) treatment in vivo. Recipient mice were sub-lethally irradiated and transplanted with leukemic BM from moribund secondary recipient mice. Three individual *Cebpa*$^{\Delta/p30}$*Tet2*$^{\Delta/\Delta}$ clones (A–C) and two *Cebpa*$^{\Delta/p30}$*Tet2*$^{+/+}$ clones (A–B) were used, respectively. The illustration was created with BioRender.com. **b** Expansion of myeloid (Mac1$^+$) donor-derived cells in bone marrow (BM) assessed by flow cytometry, and **c** *Gata2* mRNA expression in sorted leukemic blasts by qPCR assessed 24 hours after the last of three injections of 5-AZA or vehicle (samples from 3 mice per clone and 6 and 9 mice per group, for

*Cebpa*$^{\Delta/p30}$*Tet2*$^{+/+}$ and *Cebpa*$^{\Delta/p30}$*Tet2*$^{\Delta/\Delta}$, respectively). Dot plots showing individual mice for separate clones and bar graphs shows mean ± SEM for each group. Data were analyzed by Kruskal–Wallis test followed by Dunn's correction for multiple comparisons. **d** Experimental setup for evaluating the effect of 5-AZA treatment on AML progression in vivo. The illustration was created with BioRender.com. **e** Survival of sub-lethally irradiated tertiary recipient mice after transplantation of leukemic BM from moribund secondary recipient mice (clone A from both genotypes) in response to intermittent 5-AZA treatment (5-AZA treated groups 8 mice and vehicle-treated groups 4 mice). The data were analyzed by Mantel–Cox Log-rank test. Source data are provided as a Source Data file.

regulating *G2*DHE activity. Specifically, we showed that the hypermorphic effects of *CEBPA*$^{DM}$ $^{11}$, and experimental models thereof, resulted in increased *GATA2* expression compared to *CEBPA*$^{WT}$, and that CEBPA deficiency resulted in reduced *Gata2* levels. Secondly, we observed increased CEBPA binding to the *G2*DHE in *Cebpa*$^{DM}$ AML compared to normal progenitors and found that deletion or mutagenesis of the CEBPA-bound region of the enhancer resulted in lower expression of *Gata2* in *Cebpa*$^{DM}$ cells. In further support of the role of CEBPA, the *G2*DHE is highly active in *CEBPA*$^{DM}$ AML, with both elevated eRNA expression and levels of H3K27ac$^{53}$. An equally important role for CEBPA is observed in the context of inv(3) and t(3;3) AML in which inversions, translocations, and rearrangements involving the *EVI1* gene at the *MECOM* locus, lead to hijacking of the *G2*DHE to promote *EVI1* expression at the expense of *GATA2* expression thus resulting in *GATA2* haploinsufficiency$^{55–58}$. Here, *EVI1* expression was found to be downregulated following knockdown of *CEBPA* in inv(3) AML cells, and mutation of the CEBPA binding site in the hijacked enhancer reduced enhancer activity$^{58}$. In this context, *CEBPA*$^{MUT}$ would not be favorable, and these lesions are indeed underrepresented in inv(3) and t(3;3) AML$^{59–61}$.

We hypothesized that CEBPA recruits TET2 and thus mediates DNA demethylation of the *Gata2* V2 promoter in a CEBPA- and TET2-dependent manner. Indeed, we observed reduced TET2-binding to the *G2*DHE upon knockdown of *Cebpa* in *Cebpa*$^{p30/p30}$ cells. Furthermore, *Gata2* V2 levels were decreased, and *Gata2* V2 promoter DNA methylation was increased upon *Cebpa* depletion in an MLL-AF9 leukemic setting where CEBPA is dispensable for maintenance of the leukemia. The concept of CEBPA as a recruiting factor for TET2 is also supported by previous findings showing that both the p30 and p42 isoforms of CEBPA interact with TET2 via the DNA binding domain of CEBPA$^{62,63}$.

Further, CEBPA binds preferentially to methylated DNA$^{62,64}$, and has been classified as a binding site-directed DNA demethylation-inducing transcription factor$^{62,65}$. Interestingly, TET2 binds genomic regions that are enriched for CEBP motifs in myeloid cells, particularly in myeloid enhancers such as the *G2*DHE$^{26,62}$. Moreover, knockdown or knockout of *Tet2* leads to impaired upregulation of myeloid-specific genes upon *Cebpa* induction, with corresponding increased promoter methylation$^{66}$. Also, in *TET2*$^{MUT}$ or *Tet2*$^{-/-}$ leukemia an enrichment of CEBP motifs at or near hypermethylated CpGs was observed$^{26,67}$. Importantly, AML with silenced *CEBPA* is associated with DNA hypermethylation, a feature that is not present in *CEBPA*$^{DM}$ AML, which may suggest a broader function of CEBPA in the recruitment of TET2$^{68}$. In summary, we conclude that CEBPA plays an important role in the recruitment of TET2 to chromatin at the *G2*DHE, promoting DNA demethylation at the *Gata2* V2 promoter and the induction of *Gata2* expression. The extent to which this can be extended to other loci warrants further analysis but is supported by the data mentioned above.

While our findings suggest that *GATA2*$^{MUT}$ and *TET2*$^{MUT}$ both converge at rebalancing the increased expression of *GATA2* in *CEBPA*$^{DM}$ AML, patients with *CEBPA*$^{DM}$ and *GATA2*$^{MUT}$ have a more favorable prognosis$^{16,31–33,49}$ than patients harboring the *CEBPA*$^{DM}$ and *TET2*$^{MUT}$ combination$^{16,17}$. This suggests that while GATA2 deregulation plays an important role in leukemogenesis in the *CEBPA*$^{MUT}$ context, *TET2* deficiency may likely contribute to malignancy through additional mechanisms that shall remain the subject of future work. Of clinical interest, we find that TET2 deficiency renders *Cebpa*$^{DM}$ AML sensitive to 5-AZA and that TET2-deficient cells lose their proliferative advantage over TET2-proficient cells following 5-AZA treatment. In agreement with TET2-dependent *Gata2* expression, ours and previous results

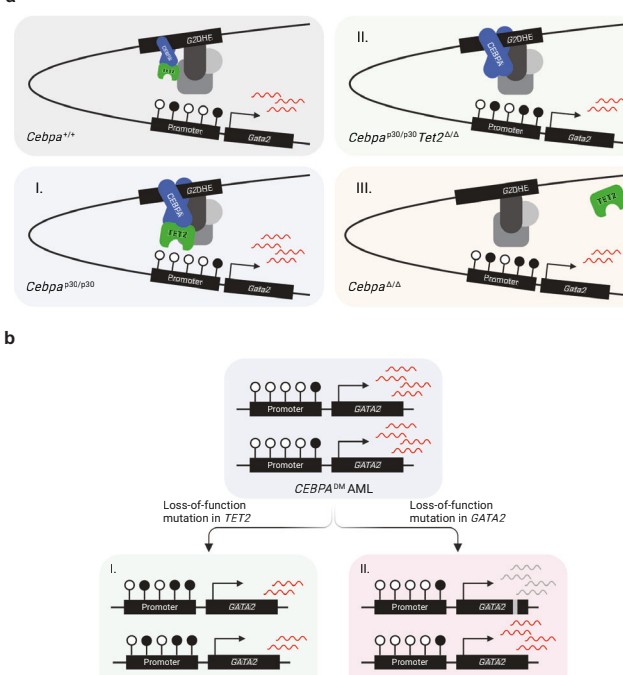

**Fig. 7 | *TET2* lesions enhance the aggressiveness of *CEBPA*-mutant AML by rebalancing *GATA2* expression. a** Model of *Gata2* differential expression as a consequence of (I) elevated CEBPA p30 due to the hypermorphic effect of the *CEBPA*[NT], (II) TET2 deficiency and, (III) CEBPA deficiency. **b** Schematic illustration of two strategies for *CEBPA*[DM] AML to rebalance *GATA2* levels by (I) loss-of-function mutations in *TET2* and (II) loss-of-function mutations in one *GATA2* allele. The illustrations were created with BioRender.com.

show that 5-AZA treatment derepresses *Gata2* expression in TET2-deficient cells[44]. Intriguingly, *CEBPA*[CT] mutations have recently been reported to sensitize AML to treatment with hypomethylating agents by disrupting the inhibitory interaction with DNMT3A mediated by the wild-type CEBPA bZIP domain[69]. Taken together, this suggests that demethylating agents could be a particularly interesting treatment option in *CEBPA*[DM]*TET2*[MUT] patients.

Finally, we note that although our mechanistic data have been acquired in experimental models of complete TET2 loss, data from AML patients indicates that *TET2* haploinsufficiency is sufficient to rebalance *GATA2* levels. We are also aware of the fact that our experimental models mimic AML in which the CEBPA p30 variant constitutes the sole CEBPA entity, which is different from the combination of N- and C-terminal mutations that constitutes the bulk of human *CEBPA*[DM] AML cases. However, since our main findings from the murine *Cebpa*[p30/p30]/*Cebpa*[Δ/p30] models are also observed in human *CEBPA*[DM] AML (including upregulation of *CEBPA* and *GATA2* in leukemic GMPs compared to normal GMPs, as well as rebalancing of *GATA2* expression and worsened outcome by the acquisition of *TET2* lesions), we believe that our observations indicate that a similar disease-relevant CEBPA-TET2 axis is active in human *CEBPA*[DM] AML.

In conclusion, our results reveal that *GATA2* is a conserved target of the *CEBPA-TET2* mutational axis in *CEBPA*[DM] AML and we propose an intricate mechanism by which elevated CEBPA p30 levels mediate recruitment of TET2 to regulatory regions of the *Gata2* gene to promote its expression. We demonstrate that increased GATA2 levels are disadvantageous to *CEBPA*[DM] leukemic cells and that this can be counteracted by TET2 loss thus providing an explanation for the co-occurrence of CEBPA and TET2 lesions in AML. Finally, increased *Gata2* promoter methylation, inflicted by TET2 deficiency, can be restored by demethylating 5-AZA treatment, thereby providing entry points for the

development of rational targeted therapies in AML patients with these mutations.

## Methods

### Patient data

**Assessment of mutational status.** To evaluate co-occurring mutations in *CEBPA*[DM] AML cases, data from published studies[3–8,17] including >40 *CEBPA*[DM] cases were extracted, and co-occurring mutations were evaluated (Supplemental Table 1). To determine frequencies of target gene mutations between *CEBPA*[DM] AML cases with *TET2*[MUT] compared to TET2 wild-type (*TET2*[WT]) AML cases, data from published studies[3–5,7,8,17,51] with specified mutational status including >40 *CEBPA*[DM] cases or corresponding cohorts were extracted and co-occurring mutations in *TET2, GATA2, WT1, CSF3R*, and *ASXL1* were evaluated (Supplemental Table 2a–e). To examine how the mutational status of *TET2* and *GATA2* were affected by *CEBPA* expression levels in AML, we utilized the publicly available data from the Beat AML cohort (Oregon Health & Science University; OHSU)[1], including 382 cases for which mutation and mRNA expression data were available. The cases were stratified based on *CEBPA* mRNA expression levels (z-score ±1.0 relative to all samples; *CEBPA*[HIGH] $n = 45$ and *CEBPA*[LOW] $n = 61$) and frequencies of *CEBPA, TET2,* and *GATA2* mutations were determined.

**Survival analysis.** The clinical data set comprises 298 patients with *CEBPA* mutations (MLL Münchner Leukämielabor GmbH), of which 152 harbored biallelic *CEBPA* mutations. Out of these 119 had specified TET2 mutational status and were included in the analyses (*CEBPA*[DM]*TET2*[WT] $n = 84$, *CEBPA*[DM]*TET2*[MUT] $n = 35$; Supplemental Table 3). All patients gave written informed consent for the use of data for scientific evaluations. The study was approved by the Internal Review Board and by the Bavarian Ethics Committee, the Bavarian State Medical Association (Bayerische Landesärztekammer) with the number 05117. The study adhered to the tenets of the Declaration of Helsinki.

**Gene expression.** The Beat AML dataset used in this study is available at http://vizome.org/aml and comprises 25 patients with *CEBPA* mutations (CEBPA[NT] and/or *CEBPA*[CT]) for which mutation and mRNA expression data is available. For the gene expression analysis, we excluded patients, which had co-occurring mutation(s) in *WT1* or *IDH1/2* since these have been shown to interfere with TET2 function[70–73] as well as two patients with low *CEBPA* variant allele frequency (VAF). Gene expression analysis was conducted on data from 16 *CEBPA*-mutant patients of which 5 have a co-occurring mutation in *TET2* (*TET2*[MUT]) (Supplemental Table 4). Differential expression analysis was performed with DESeq2[74] (v. 1.26.0, RRID:SCR_015687) and default parameters. To assess gene expression changes in *CEBPA*[WT] patients with *TET2*[MUT] vs *TET2*[WT], we included patients with normal karyotype AML from the Beat AML dataset and excluded patients with mutation(s) in *WT1* or *IDH1/2* (*CEBPA*[WT]*TET2*[MUT] $n = 34$ and *CEBPA*[WT]*TET2*[WT] $n = 167$). To analyze *CEBPA* and *GATA2* expression levels in *CEBPA*[WT] vs. *CEBPA*[NT] AML mRNA expression of the two genes together with mutational and karyotype status data was retrieved from the Beat AML study[1] via cBioPortal[75,76] (RRID:SCR_014555). We evaluated patients for whom data was available for genomic profiling including mRNA expression, mutations, and karyotype. We included patients with normal karyotype AML and excluded patients with mutation(s) in *WT1* or *IDH1/2* (*CEBPA*[WT] $n = 52$ and *CEBPA*[NT] $n = 15$).

### In vitro experiments

**Competitive CRISPR-targeting.** For generation of *Tet2* or *Gata2* mutated clones, *Cebpa*[p30/p30] (♂) cells[37] were electroporated with ribonucleoparticles containing recombinant Cas9 nuclease from Streptococcus pyogenes (Sp) (#1081058, IDT), tracrRNA (#1075927, IDT) and crRNAs (Alt-R® CRISPR-Cas9 crRNA, IDT) targeting *Tet2* and *Gata2*, respectively. crRNAs were designed using the CHOPCHOP[77]

web tool (chopchop.cbu.uib.no, RRID:SCR_015723) (Supplemental Table 5). crRNA and tracrRNA molecules were complexed at room temperature and assembled with recombinant SpCas9 according to the manufacturer's protocols (IDT). Pools of *Tet2*- or *Gata2*-targeted cells were screened at regular intervals to monitor the outgrowth of subpopulations. The genomic regions that were targeted with CRISPR/Cas9 technology were PCR-amplified, Sanger sequenced, and analyzed with the online tool Tracking of Indels by DEcomposition (TIDE)[78] for insertions or deletions (indels) in the targeted region. Primers for PCR are provided in Supplemental Table 6.

**Gata2 enhancer CRISPR-targeting.** sgRNA sequences targeting the *Gata2* distal hematopoietic enhancer (G2DHE) were obtained from the UCSC Genome Browser[79] (genome.ucsc.edu, RRID:SCR_005780) and targets with a high predicted cleavage (Doench/Fusi 2016 Efficiency > 55)[80] selected (Supplemental table 5). SpCas9-expressing *Cebpa*^p30/p30*Tet2*^MUT cells were isolated after lentiviral expression of lenti-Cas9-Blast (#52962 Addgene). *Cebpa*^p30/p30 and *Cebpa*^p30/p30*Tet2*^MUT cells were co-transduced with pLenti-hU6-sgG2DHE_A/B-IT-PGK-iRFP and LentiGuide-sgG2DHE_1–6-Puro-IRES-GFP. GFP+iRFP670+ cells were sorted via fluorescence-activated cell sorting (FACS) and frozen for subsequent analysis.

**Cebpa shRNA-knockdown.** *Cebpa*^p30/p30 rtTA3 cells expressing Dox-inducible shRNAs against Renilla luciferase (shRen, control) or *Cebpa* (sh*Cebpa*) were used as previously described[12]. *Cebpa* knockdown was induced by the addition of 4-hydroxytamoxifen (4-OHT; 1 μM; #H7904 Sigma-Aldrich) to the cell culture medium to activate shRNA expression (mean *Cebpa* knockdown efficiency > 90% compared to shRen control) and cells were collected for further analysis after 48 h.

### In vivo experiments

Experiments were carried out according to protocols approved by the Danish Animal Ethical Committee. Mice were bred and housed locally at the Department of Experimental Medicine at the University of Copenhagen. The mice were housed in a temperature- and humidity-controlled room with a 06:00–18:00 h light cycle and fed a standard chow diet and tap water *ad libitum*. We used *Tet2*^fl/fl [81], *Cebpa*^p30/+ [13], *Cebpa*^+/fl [82] and *Mx1-Cre*^+ [83] lines to generate inducible *Tet2*-deficient and *Cebpa*-mutant compound lines. The following genotypes were used for experiments: *Cebpa*^fl/p30*Tet2*^+/+, *Cebpa*^+/fl*Tet2*^fl/fl, *Cebpa*^fl/p30*Tet2*^fl/fl, *Cebpa*^fl/p30*Tet2*^+/+*Mx1-Cre*^+, *Cebpa*^+/fl*Tet2*^fl/fl*Mx1-Cre*^+, and *Cebpa*^fl/p30*Tet2*^fl/fl*Mx1-Cre*^+. *Cebpa*^p30/p30 embryos were generated as previously described[13]. We used *iMLL-AF9*^+ [84], *Cebpa*^+/fl [82] and *R26-CreER*^+ [85] lines to generate an *iMLL-AF9*^+*Cebpa*^fl/fl*R26-CreER*^+ compound line. Primers for genotyping in are provided in Supplemental Table 6.

During the leukemia initiation and propagation experiments described below, the animals were monitored daily and euthanized when they showed sign(s) of sickness e.g., inactivity, hunched posture, white paws, and/or matted or puffed-up fur as well as pain assessed based on the NC3R's mouse grimace scale[86] or reduced bodyweight (maximal allowed reduction = 15%). The experiments were terminated after 12 months.

**Leukemia initiation (Cebpa^Δ/p30 model).** C57BL/6 J.SJL congenic recipients (female, 10–12 weeks old) were lethally irradiated (900 cGy) 12–24 h prior to intravenous injection with $1 \times 10^6$ bone marrow (BM) cells from individual donor mice. The mice were given Ciprofloxacin (100 mg/l in acidified water; #17850 Sigma-Aldrich) in the drinking water to prevent infections 3 weeks post-irradiation. Recipient mice were allowed to recover for 6 weeks post-transplantation before Cre-LoxP recombination was induced by two intraperitoneal injections of Poly(I)·Poly(C) (300 μg in 200 μl PBS; #27-4732-01 GE Healthcare) with 48 h rest in-between. The day of the first injection was set as time-point zero for the survival study and mice were monitored for leukemia

development and euthanized when moribund. To follow leukemia initiation in the recipients, a subgroup of mice was subjected to blood and BM sampling at 12, 24, and 36-week time-points. BM from moribund mice was collected and frozen (10% DMSO in FBS; #D8418 Sigma-Aldrich, #HYCLSV30160.03 Hyclone) for subsequent FACS and analysis.

**Leukemia initiation (Cebpa^p30/p30 model).** C57BL/6 J.SJL congenic recipients (female, 10 weeks old) were lethally irradiated (900 cGy) 12–24 h prior to intravenous injection with $0.5$-$1 \times 10^6$ fetal liver cells from E15.5 *Cebpa*^p30/p30 embryos. The mice were given Ciprofloxacin (100 mg/l in acidified water) in the drinking water to prevent infections 3 weeks post-irradiation. Latency to leukemic initiation was 8-11 months.

**Leukemia propagation.** C57BL6/6 J.SJL recipients (female, 10–12 weeks old) were lethally irradiated (900 cGy) 12–24 h prior to being intravenously injected with $2 \times 10^5$ thawed live BM cells from moribund donor mice together with $4 \times 10^5$ freshly isolated BM cells from C57BL6/6 J.SJL mice. The day of the injection was set as time-point zero for the survival study and mice were monitored and euthanized when moribund. The mice were given Ciprofloxacin in the drinking water to prevent infections 3 weeks post-irradiation.

**Competitive shRNA-knockdown.** C57BL/6 J.SJL recipients (female, 10–12 weeks old) were sub-lethally irradiated (500 cGy) 12–24 h prior to being intravenously injected with a 1:1 mix of *Cebpa*^p30/p30 cells[13] transduced with shRNA targeting *Gata2* (detailed in ShRNA knockdown below) or with control-shRNA[87]. The ratio of *Gata2*- or control-shRNA-GFP+ to control-shRNA-YFP+ cells was analyzed by flow cytometry four weeks later.

**5-azacytidine treatment.** C57BL/6 J.SJL recipients (female, 10–12 weeks old) were sub-lethally irradiated (500 cGy) 12–24 h prior to being intravenously injected with $1 \times 10^5$ thawed live BM cells from moribund secondary recipient mice. The mice were given Ciprofloxacin in the drinking water to prevent infections 3 weeks post-irradiation. The mice received intraperitoneal injections with the demethylating agent 5-azacytidine (2.5 mg/kg/day in saline; #A2385 Sigma-Aldrich) at days 6–10 and 20–24 post-transplantation. The time of the BM cell injection was set as time-point zero for the survival study and mice were monitored and euthanized when moribund. To evaluate the effects of short-term 5-azacytidine treatment, recipient mice were treated at days 13–15 and euthanized 24 h after the last injection. BM was collected for FACS, and sorted cells were frozen for subsequent analysis.

### Ex vivo cell culture

**Establishment of ex vivo Cebpa^Δ/p30Tet2^+/+ and Cebpa^Δ/p30Tet2^Δ/Δ lines.** Thawed cryo-preserved cells from primary AML were cultured in Lonza X-Vivo™ 15 cell medium (#BE02-060Q Thermo Fisher Scientific) supplemented with Bovine Serum Albumin in Iscove's MDM (10%; #09300 Stemcell™ Technologies), Penicillin-Streptomycin (1%; #15140122 Gibco), β-mercaptoethanol (0.1 mM; #31350010 Gibco), L-glutamine (2 mM; #25030149 Gibco), and cytokines h-IL-6 (50 ng/ml; #130-093-032 Miltenyi Biotec), m-SCF (50 ng/μl; #250-03 Peprotech), m-IL-3 (10 ng/ml; #213-13 Peprotech), and m-GM-CSF (10 ng/ml; #315-03 Peprotech). Two clones of each genotype (*Cebpa*^Δ/p30*Tet2*^+/+ and *Cebpa*^Δ/p30*Tet2*^Δ/Δ) continued to expand beyond 40 days and withstood freeze-thawing, and these clones have been used for further experiments.

**Vitamin C treatment.** Cells were seeded at a density of $2 \times 10^5$ cells/ml and the cell culture medium was supplemented with vitamin C (100 μg/ml; L-ascorbic acid, #A8960 Sigma Aldrich). Live cells were

counted using Solution 13 (AO-DAPI; #910-3013 Chemometec) on a NucleoCounter® NC-3000™ and reseeded at $2 \times 10^5$ cells/ml every third day. The experiments were run with a total of two biological replicates per genotype (performed on separate days) where each experiment assayed one leukemic line per genotype. Each condition (Vitamin C and vehicle) was performed in technical triplicates for each of the two biological replicates per genotype.

**5-azacytidine treatment.** Cells were seeded at a density of $2 \times 10^5$ cells/ml and medium supplemented with 5-azacytidine (5-AZA; 1 µg/ml; #A2385 Sigma-Aldrich). Live cells were counted using Solution 13 on a NucleoCounter® NC-3000™ and reseeded at $2 \times 10^5$ cells/ml days three and six. 24 hours later, up to $1 \times 10^5$ cells were isolated and resuspended in RA1 buffer (NucleoSpin RNA XS, #740902 Macherey-Nagel). The experiments were run with a total of two biological replicates per genotype (performed on separate days) where each experiment assayed one leukemic line per genotype. Each condition (5-AZA and vehicle) was performed in technical triplicates for each of the two biological replicates per genotype.

**Establishment of ex vivo iMLL-AF9⁺Cebpa^{fl/fl}R26-CreER⁺ lines.** Sorted GMPs from *iMLL-AF9⁺Cebpa^{fl/fl}R26CreER⁺* mice, were cultured in MethoCult (M3434; #03434, Stemcell technologies) supplemented with doxycycline (1 µg/ml; #D9891 Sigma-Aldrich) for three replatings to induce expression of the MLL-fusion protein.

**Cebpa knockout.** Leukemic *iMLL-AF9⁺Cebpa^{fl/fl}R26CreER⁺* cells were cultured in RPMI 1640 medium (#21875034, Gibco) supplemented with FBS (10%), Penicillin-Streptomycin (1%), doxycycline (1 µg/ml), and cytokines m-IL-3 (6 ng/ml), m-SCF (50 ng/ml), and h-IL-6 (10 ng/ml). After two days, 4-hydroxytamoxifen (4-OHT; 1 µM; #H7904 Sigma-Aldrich) or vehicle was added to the cell culture medium to activate *Cre-LoxP* recombination, resulting in a reduction of *Cebpa* mRNA to $1.7 \pm 0.3\%$ vs. $100 \pm 12.3\%$ in vehicle control. Three days later cells were isolated and either frozen or resuspended in RA1 buffer (NucleoSpin RNA XS, # 740902 Macherey-Nagel). The experiments were run with a total of two biological replicates (performed on separate days). Each condition (4-OHT and vehicle) was performed in 2-3 technical triplicates for each of the two biological replicates.

## ShRNA knockdown

**Cloning of shRNA into pMLS vector.** Murine shRNAs targeting *Gata2* (sh*Gata2*) were cloned into MSCV-LTRmir30-SV40-GFP vector. Targeting sequences were identified from the Mission® shRNA library (Supplemental Table 7) and the sense and anti-sense sequences were incorporated with a miR-30-loop to generate a 97-mer target sequence. Oligonucleotides were amplified by PCR using miR30 common primers (Supplemental Table 6), which include restriction sites for *XhoI* and *EcoRI*. The resulting 138-mer PCR amplicons and the vector were digested with *XhoI* and *EcoRI* and products were ligated using T4 DNA Ligase (#15224025 Invitrogen). Bacterial transformation was performed to amplify individual ligation products, and correct inserts were verified by Sanger Sequencing. These, together with vectors containing a control non-targeting sequence (MSCV-LTRmir30-SV40-GFP and MSCV-LTRmir30-SV40-YFP), were used in subsequent transfection/transduction experiments, as previously described[87,88].

**Transduction of Cebpa^{p30/p30} cells.** Retroviral transduction was done as previously described[87]. Briefly, retroviral supernatants were generated by transfection of Phoenix-Eco cells (RRID:CVCL_H717). For transduction, retroviral supernatant was added onto retronectin-coated (1:25; #T100B TaKaRa) non-tissue culture treated plates and centrifuged at 2000 ×g for 60 min at 4 °C. After aspiration of the supernatant, *Cebpa^{p30/p30}* cells were seeded at a density of $0.5-1 \times 10^5$

cells/cm². The transduction was repeated the following day, and the cells were cultured for 24 h prior to FACS sorting of transduced (GFP⁺/YFP⁺) cells on a BD FACSAria™ III (BD Bioscience). The efficiency of shRNA-mediated gene expression knockdown was assessed with qPCR and cells were used for transplantation and assessment of their competitiveness in vivo.

## Immuno-staining

*Flow cytometry.* To analyze the composition of either freshly isolated or thawed cryopreserved BM and blood, cells were stained with fluorescently labelled antibodies. For blood analysis, 50 µl blood was collected from the facial vein and erythrocytes were lysed with lysing buffer (BD Pharm Lyse™, #555899 BD Bioscience). For BM analysis, cells were collected by crushing tibia, femur, and ilium and filtered through a 50 µm filcon cup (#340630 BD Bioscience). Blood or BM cells were washed in PBS with 3% FBS and stained with fluorescently labelled antibodies for 30 min at 4 °C (Supplemental Table 8). For cryopreserved cells, the cells were counterstained with DAPI (1:10000; #D3571 Invitrogen) to separate out dead cells. Fluorochrome-minus-one was used as controls. Flow cytometry data was obtained using a BD FACSAria™ III or a BD LSR II™ (BD Bioscience) and analyzed using FlowJo software (v9, RRID:SCR_008520).

For downstream transcriptional and epigenetic analyses, live donor-derived non-lymphoid and non-erythroid cells (DAPI⁻CD45.2⁺CD3⁻B220⁻Ter119⁻) were sorted using a BD FACSAria™ III, spun down and cell pellets were either snap-frozen or resuspended in RLT buffer (RNeasy Mini Kit, #74104 Qiagen).

For ex vivo cell culture of *iMLL-AF9⁺Cebpa^{fl/fl}R26-CreER⁺* cells, c-kit⁺ BM cells were enriched by magnetic sorting (mouse CD117 MicroBeads; #130-091-224, Miltenyi Biotec), and granulocyte/monocyte progenitors (GMPs; Lin⁻C-kit⁺Sca1⁻CD41⁻FcgRII⁺) were sorted using a BD FACSAria™ III.

**Immunohistochemistry.** To evaluate the proliferative status of leukemia cells, cells from BM of moribund mice were spun on glass slides, air-dried, and fixed with methanol (#VWRC20846.292 VWR). After blocking of endogenous peroxidase activity with hydrogen peroxide (1%), slides were stained overnight at 4 °C with anti-Ki67 antibody (1:50; Clone SP6, RRID:AB_302459, #ab16667 Abcam) in antibody diluent (S3022 Dako). To visualize the primary antibody, EnVision HRP Rabbit (K4003 Dako) together with Vina Green™ Chromogen Kit (BRR807 Biocare Medical) was utilized according to manufacturer's instructions. The cells were counterstained with Mayer Hematoxylin (#51275 Sigma-Aldrich), dehydrated and coverslips mounted with Entellan (#107960 Sigma-Aldrich). Images were captured using a Leica microscope at 20X magnification and Ki67⁺ cells were quantified out of one hundred cells.

**Western blotting.** Western blotting for TET2 was performed according to standard laboratory protocols, using the following antibodies: anti-TET2 (1:100, Clone C-7, RRID:AB_2924805, #sc-398535 Santa Cruz) and anti-HSC70 (1:10000, Clone B-6, RRID:AB_627761, #sc-7298 Santa Cruz).

## Quantitative PCR

RNA from sorted blasts or ex vivo-cultured cells was isolated using NucleoSpin RNA XS kit (#740902 Macherey-Nagel) or RNeasy Mini Kit (#74104 Qiagen) according to the manufacturers' instructions and converted to cDNA using ProtoScript First Strand cDNA Synthesis Kit (#E6300 New England BioLabs). Quantitative PCR (qPCR) to assess knockdown efficiency was run using TaqMan Fast Advanced Master Mix (#4444556 Applied Biosystems) and TaqMan assay for *Gata2* (Mm00492301_m1 FAM-MGB), in duplex with housekeeping gene *18 S* (Hs99999901_s1 VIC-MGB-PL). TaqMan assay for *Ki67* (Mm01278617_m1 FAM-MGB) was used to assess the expression of the

proliferation marker. qPCR to evaluate mRNA levels of total *Gata2*, *variant 1* (*V1*) and *variant 2* (*V2*), respectively, was run in duplex using LightCycler 480 SYBR Green I Master (#04887352001 Roche) with primers for *Gata2* and housekeeping gene *Actb* and *Gapdh*[41] (Supplemental Table 6). Gene expression was calculated with the $2^{-\Delta\Delta ct}$ method.

RNA from *Cebpa*[p30/p30] cell lines was isolated using RNeasy Plus Mini Kit (#74134 Qiagen) according to the manufacturer's instructions and converted to cDNA with RevertAid First Strand cDNA Synthesis Kit (#K1622 Thermo Scientific). qPCR was run using SsoAdvanced Univ SYBR Grn Suprmix (#1725271, Bio-Rad Laboratories Ges.m.b.H.) and primers for *Gata2* and *Gapdh* (Supplemental Table 6).

## Bisulfite PCR

DNA was isolated using DNeasy Blood and tissue kit (#69504 Qiagen) and the DNA was bisulfite converted using EZ-DNA Methylation Gold Kit (#D5005 Zymo Research), both according to the manufacturer's instructions. PCR was run using Pfu Turbo Cx Hotstart DNA polymerase (#600410 Agilent) with primers targeting a part of the CpG island in the *Gata2 V2* promoter region (Supplemental table 6). After verification of their correct size, PCR products were cloned using Zero Blunt Topo PCR Cloning kit (#450245 Invitrogen), and single colonies were picked and amplified. Plasmid DNA was isolated using NucleoSpin Plasmid EasyPure (#740727.250 Macherey-Nagel), the correct insert size was verified after cleavage with restriction enzyme EcoRI (#R0101 New England Biolabs) and sent for Sanger sequencing using the M13 primer provided with the cloning kit.

## Chromatin immunoprecipitation (ChIP)-qPCR

ChIP for CEBPA was performed as previously described[37] using an anti-CEBPA antibody (1:60, C-18, RRID:AB_2078046, #sc-9314, Santa Cruz Biotechnology). ChIP for TET2 was performed using an anti-TET2 antibody (1:50; clone D6C7K, RRID:AB_2799102, #36449, Cell Signaling Technology), as previously described[37], including a 30-minute incubation with 2 mM disuccunumidyl glutarate (DSG; #20593 Thermo Scientific) before the 1% formaldehyde crosslinking step. The sequences used for qPCR are listed in Supplemental table 6.

## High-throughput sequencing and bioinformatic analyses

**RNA-sequencing (RNA-seq) of cell line models.** RNA was isolated from $1 \times 10^6$ cells using RNeasy Plus Mini Kit (#74134 Qiagen) according to the manufacturer's instructions and quality was assessed on a Bioanalyzer 2100 G2939A (Agilent). 1 µg of RNA was used to generate sequencing libraries using QuantSeq 3′ mRNA-Seq Library Prep Kit (FWD) for Illumina, 96 preps (#015.96, Lexogen) and the PCR Add-on Kit for Illumina, 96 rxn (#020.96, Lexogen). The libraries were quantified on a Bioanalyzer 2100 G2939 (Agilent) and pooled in equimolar amounts. Multiplexed libraries were sequenced on a HiSeq4A (Illumina).

**RNA-seq of leukemic cells from in vivo models.** RNA was isolated from $5 \times 10^5$ sorted cells using RNeasy Mini Kit (#74104 Qiagen) according to the manufacturer's instructions and quality was assessed by RNA 6000 Pico Kit (#5067-1513 Agilent) on a Bioanalyzer 2100 (Agilent). 200 ng RNA was used to generate sequencing libraries using TruSeq RNA Library Prep Kit v2 (#RS-122-2001 Illumina). The libraries were quantified using Qubit dsDNA BR Assay Kit (#32853 Thermo Fisher Scientific) and DNA 1000 Kit (#5067-1504 Agilent) and pooled in equimolar amounts. Multiplexed libraries were sequenced on a NextSeq 500 (Illumina) using NextSeq 500 High Output v2 Kit (75 cycles; #FC-404-2005 Illumina).

**Bioinformatics analyses of RNA-seq data.** RNA-seq analysis for in vitro *Cebpa*[p30/p30] cells was performed as previously described[12,37]. Quality check was done with FastQC[89] (v. 0.11.4, RRID:SCR_014583) and

preprocessing with PRINSEQ-lite[90] (v. 0.20.4; RRID:SCR_005454), using parameters: -min_len 30 -min_qual_mean 30 -ns_max_n 5 -trim_tail_right 8 -trim_tail_left 8 -trim_qual_right 30 -trim_qual_left 30 -trim_qual_window 5. The remaining reads were aligned against the mouse reference genome (mm10) with BWA[91] (v. 0.7.15; RRID:SCR_010910). RNA-seq analysis for in vivo *Cebpa*[Δ/p30] cells was performed as follows. RNA-seq reads were processed with the bcbio RNA-seq pipeline[92] (https://github.com/bcbio/bcbio-nextgen, RRID:SCR_004316) and the bcbioRNASeq R package (https://github.com/hbc/bcbioRNASeq). In brief, transcript abundance estimates were obtained using Salmon[93] (v. 0.12.0, RRID:SCR_017036) against reference transcriptome GRCm38/mm10 ENSEMBL release 94, summarized to gene level and imported into R using tximport[94] (v. 1.10.1, RRID:SCR_016752) (using setting countsFromAbundance = "lengthScaledTPM"). Differential gene expression analysis between the *Cebpa*[Δ/p30]*Tet2*[+/+] and *Cebpa*[Δ/p30]*Tet2*[Δ/Δ] genotype was performed using DESeq2 with standard parameters[74] (v. 1.22.2, RRID:SCR_015687) excluding lowly expressed genes (< 10 sum normalized counts across all samples) and running with alpha = 0.05.

Gene expression levels between primary *Cebpa*[Δ/p30]*Tet2*[+/+] and established *Cebpa*[p30/p30] leukemias were compared using edgeR (v. 3.32.1, RRID:SCR_012802).

**Gene set enrichment analysis (GSEA).** GSEA was performed using the GSEA software[95,96] (v. 4.1.0, RRID:SCR_003199) and the Molecular Signatures Database (RRID:SCR_016863).

**Assay for transposase-accessible chromatin-sequencing (ATAC-seq).** ATAC-seq was performed as previously described[12].

**Bioinformatics analyses of ATAC-seq data.** Analysis of ATAC-seq was performed as previously described[12]. HOMER[97] (v. 4.11, RRID:SCR_010881) was used to identify motifs enriched in the ATAC peaks.

**Bisulfite whole genome sequencing (WGBS).** DNA was isolated from $1 \times 10^6$ sorted cells using DNeasy Blood and tissue kit (#69504 Qiagen) according to the manufacturer's instructions. Bisulfite conversion of DNA was done according to manufacturers' instructions using EZ-DNA Methylation Gold Kit (#D5005 Zymo Research). Quality of bisulfite treated DNA was assessed by RNA 6000 Pico Kit (#5067-1513 Agilent) on a Bioanalyzer 2100. Libraries of bisulfite-converted DNA were prepared using Pico Methyl-Seq Library Prep Kit (#D5455 Zymo Research) according to manufacturer's instructions and the final concentration and quality of the libraries was assessed using Qubit dsDNA HS Assay Kit (#Q32854 Thermo Fisher Scientific) and High Sensitivity DNA Analysis Kit (#5067-4626 Agilent) on a Bioanalyzer. Duplexed libraries were sequenced on a NextSeq 500 (Illumina) using NextSeq 500 High Output v2 Kit (75 cycles).

**Bioinformatics analyses of WGBS data.** Reads were trimmed and filtered using Trim Galore[98] (v. 0.4.3, RRID:SCR_011847) with default parameters, and quality was assessed before and after using FastQC[89] (v. 0.11.7). Trimmed reads were aligned to the mouse genome assembly GRCm38 (mm10) using Bismark[99] (v. 0.19.1, RRID:SCR_005604) with option -non_directional (other parameter left at default values; this used Bowtie 2[100] (v. 2.2.8 RRID:SCR_016368) with -q --score-min L,0,−0.2 --ignore-quals). After deduplication of alignments (using deduplicate_bismark), the methylation information for individual cytosines was extracted using bismark_methylation_extractor (--cytosine_report --comprehensive --gzip). To quantify DNA methylation of gene bodies and promoters (1000 bp up-and downstream of transcription start sites), we used the weighted methylation level (i.e., summarizing over all CpG positions in the given region, the number of reads supporting methylated cytosine divided by the number of all reads covering these positions). Plots of average methylation levels

across extended gene bodies were generated using deepTools[101] (v.3.1.3, RRID:SCR_016366) computeMatrix (scale-regions -m 4000 -a 1000 -b 1000 --unscaled5prime 1000 --unscaled3prime 1000) and plotProfile, for which Bismark-generated bedGraph files were converted to BigWig format (using UCSC's bedGraphToBigWig[102] (v. 4)).

**Bioinformatic analyses of chromatin immunoprecipitation-sequencing (ChIP-seq).** ChIP-seq data from in vitro *Cebpa*[p30/p30] cells was processed as described[12]. ChIP-seq data from in vivo *Cebpa*[p30/p30] cells was processed as follows; raw reads derived from CEBPA (*Cebpa*[+/+] and *Cebpa*[p30/p30]) ChIP-seq experiments were mapped to mouse (mm10) genome assembly using Bowtie 2[100] (v. 2.3.4.3). We used uniquely mapped and PCR duplicates (exact copies) collapsed as one read and extended to their fragment length by determining the read extension size using MACS2[103] (v. 2.1.0.20151222; predicted parameter, RRID:SCR_013291). Raw read counts were normalized to TPM using deepTools[101] (v. 3.3.1; bamCoverage). Raw read counts (CEBPA binding levels) mapping to *Gata2* promoter and enhancer regions were computed using bedtools[104] (v. 2.30.0; multicov, RRID:SCR_006646), and the differences in CEBPA binding between *Cebpa*[+/+] and *Cebpa*[p30/p30] conditions were computed using DESeq2[74] (v. 1.30.1). Sequencing reads derived from TET2 ChIP-seq experiment were preprocessed with PRINSEQ-lite[90] (v. 0.20.4; RRID:SCR_005454) and the remaining reads were mapped to the mouse reference genome sequence (mm10) using BWA[91] (v. 0.7.17-r1188, RRID:SCR_010910). The resulting alignments were processed with samtools[105] (v. 1.13; RRID:SCR_002105) and peak calling was done with MACS2[103] (v. 2.1.0.20140616; RRID:SCR_013291). Aligned read counts were normalized to RPKM using the bamCoverage function from deeptools[101] (v. 3.5.1; RRID:SCR_016366).

## Statistics

Data were analyzed for significance using parametric tests, with prior log-transformation if necessary to achieve normal distribution. Normality was evaluated by Shapiro–Wilk test. Two-group analyses were done using an unpaired two-tailed $t$-test. Multiple-group analyses were done with one-way-ANOVA followed by multiple comparisons correction using Dunnett when comparing to a reference group, or two-way-ANOVA followed by multiple comparisons correction using Šídák test when comparing two independent factors across four groups. Data sets that did not pass normality tests were analyzed by Kruskal–Wallis test followed by multiple comparisons correction using Dunn's test. Survival curves were analyzed using Mantel–Cox Log-rank test. To compare distributions Wilson/Brown binominal test was used. To compare a median against a hypothetical median Wilcoxon signed-rank test was used. $p$-values < 0.05 were considered statistically significant. Data was analyzed using GraphPad Prism (v. 9, RRID:SCR_002798). Data is shown as mean ± SEM unless otherwise stated.

## Reporting summary

Further information on research design is available in the Nature Portfolio Reporting Summary linked to this article.

## Data availability

The data generated in this study is publicly available in Gene Expression Omnibus (GEO) under accession numbers GSE214224 (RNA-seq, ATAC-seq, and TET2 ChIP-seq in vitro) and GSE213864 (RNA-seq and WGBS in vivo), and within the article and its supplementary files. The following other publicly available data was used in this study: CEBPA and H3K27Ac ChIP-seq from myeloid progenitor cell model for p30-driven AML[12] is available under GSE158727. CEBPA ChIP from mouse *Cebpa*[+/+] or *Cebpa*[p30/p30] GMPs[11] is available under GSE118963. RNA-seq data from *Cebpa*[p30/p30] AML[11,34] are available under GSE118963 and GSE141477. Patient data analyzed in this study were from the Beat AML study (accessed through cBioPortal[75,76] [https://www.cbioportal.org/]

or Vizome[1] [http://www.vizome.org/]) or from published cohort studies (Supplemental tables 1, 2a–e). Source data are provided with this paper.

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

## Acknowledgements

Work in the Porse lab was supported by grants from the Copenhagen University Hospital, the Capital Region of Copenhagen, the Independent Research Fund Denmark (9039-00189B) and through a center grant from the Novo Nordisk Foundation (Novo Nordisk Foundation Center for Stem Cell Biology, DanStem; Grant Number NNF17CC0027852). This work has been performed in the context of the Danish Research Center for Precision Medicine in Blood Cancers funded by the Danish Cancer Society (R223-A13071) and Greater Copenhagen Health Science Part-ners. Work in the Grebien lab was supported by the European Union's Horizon 2020 research and innovation program (European Research

Council grant agreement No 636855 and Austrian Science Fund (FWF), grants no. TAI-490 and P35628. Work in the Schoof lab was supported by a grant from the Independent Research Fund Denmark (case no. 2067-00053B). ASW was supported by grants from The Lundbeck Foundation (R303-2018-2868) and The Swedish Research Council (2015-00517). EH was supported by a grant from Forschungsförderung from the Fellinger Krebsforschungsverein. We thank members of the Grebien and Porse laboratories for their discussions. We thank Anna Fossum for her excellent research assistance.

## Author contributions

E.H., A.S.W., G.M., M.B.S., E.R., T.D.A., A.K.F., C.G., J.W., and E.M.S. performed experiments. E.H., AS.W., A.W., T.E., M.B.S., S.P., and E.M.S. analyzed data. E.H., A.S.W., F.G., and B.T.P. designed experiments. J.F. contributed essential material. M.M. and T.H. provided clinical data. E.H., A.S.W., F.G., and B.T.P. drafted the manuscript. All authors have proofread and approved the final version of the manuscript.

## Competing interests

The authors declare no competing interests.
