## [Peer Review File · Nature Communications]

REVIEWER COMMENTS

Reviewer #1 (Remarks to the Author):

The authors study cooperation between TET2 and CEBPA mutations in AML by analyzing published human patient datasets and experimental modeling in mice and with murine cells in vitro.

They show that synchronous deletion of Tet2 and Cebpa p42 results in an accelerated leukemia phenotype in mice as compared to either mutation alone. Comparing human datasets with in vitro and in vivo mouse data, they find that Fut8, Gata2, and Sirt5 are conserved (across species and in these different experimental setups) as targets downregulated upon loss of Tet2. Among these targets, only the Gata2 V2 promoter shows changes in promoter methylation. The authors then show that 25-60% downregulation by RNAi (but not >75% downregulation) and heterozygous deletion by CRISPR/Cas9 (but not homozygous deletion) confers a competitive advantage in Cebpa p42-deficient cells. Meanwhile, the Gata2 -77 kb enhancer is bound by Cebpa p30 and this binding is associated with increased Gata2 expression, leading the authors to conclude that Tet2 co-mutations "rebalance" Gata2 levels deranged by Cebpa mutations. Finally, azacitidine ex vivo restores Gata2 expression levels on this genomic background and improves survival.

Overall, the text and figures are well designed and described, and the story is clear and quite comprehensive. Some issues require further clarification.

1. My major comment has to do with the data flitting between somewhat different models to synthesize the results presented: When used at its best (as in Fig. 3a) it can help to narrow targets down to a small handful of conserved ones. However, in Fig. 2 (as an example), it is not clear why the authors move between Cebpa(p30/p30) cells in vitro and Cebpa(Δ /p30) mice [*], study Tet2(-/-) effects in the mice to model human TET2 mutations (presumably heterozygous), and compare Cas9 targeting in a myeloid progenitor cell pool in vitro with transformed blasts in vivo.

While in some respects it is not uncommon to use Tet2(-/-) mice in such studies, the discrepancy becomes somewhat more salient in this case where the finding is that Tet2 mediates a change in Gata2 expression promoting leukemogenesis with 25-60% downregulation but not with >75% downregulation.

[*] The authors call this Cebpa(-/p30), but it is an Mx1-Cre inducible deletion after transplant.

Additional specific comments

p. 7, line 33: "concomitant mutations in both genes would be redundant and thus . . . mutually exclusive" — one would think that it is synthetic lethality that would explain mutual exclusivity; it's not clear that this line of argumentation here is really necessary: based on observational human data, there is a slight paucity of co-occurring Tet2 and Gata2 mutations, but the mechanistic work isn't being undertaken to distinguish redundancy from synthetic lethality.

p. 8, line 23: In what cell type is this experiment undertaken? Neither the text nor figure legend makes this clear. Do the results differ in different cell types?

p. 8, lines 28-30: In this specific experiment with CRISPR/Cas9-mediated disruption of the Gata2 enhancer element bound by Cebpa, is abrogation of binding to that enhancer element associated with decreased Gata2 V2 expression (as opposed to calling back to a distinct experiment in Fig. 3F-G)?

p. 9, lines 12-28; Fig. 6: The relative expression of Gata2 mRNA is decreased to nearly zero in Tet2-deficient mice (without azacitidine treatment): yet earlier in the work it is said that efficient downregulation of Gata2 expression (>75%) does not provide a competitive advantage and, in fact, homozygous deletion of Gata2 is not viable. How would the authors explain the apparent contradiction here? Is the analysis confounded by a higher percentage of dying cells, more rapid cell cycling, different cell maturation stage at this time point?

Reviewer #2 (Remarks to the Author):

Mutational cooperativity is a critical part of leukemogenesis, and Heyes et al. sought to illuminate the mechanistic basis for the cooperation between mutations in CEBPA and TET2 in acute myeloid leukemia (AML). The CEBPA gene is biallelically mutated (double mutated; CEBPA-DM) in a subset of AML patients that harbor either biallelic CEBPA N-terminal (NT) mutations or a combination of a NT mutation together with a C-terminal (CT) mutation in the other allele. CEBPA NT lesions promote the expression of a truncated hypermorphic p30 isoform, whereas CEBPA CT mutations result in amorphic CEBPA variants unable to dimerize or bind DNA. To study the impact of double mutant CEBPA in AML, the authors used cell and murine models with a CEBPA-/p30 and CEBPA p30/p30 genetic background. The former presumably mimicking the double mutant CEBPA with a NT and CT mutation, while the latter mimicking the biallelic CEBPA NT mutations. Next, to understand the impact of double mutant CEBPA AML and co-occurring lesions with TET2, the authors introduced TET2 mutations with CRISPR or by conditional knockout of the Tet2 alleles. The authors described that TET2 loss-of-function in double mutant CEBPA AML led to an aggressive disease phenotype by rebalancing the increased and suboptimal levels of GATA2 that are induced by hypermorphic CEBPA NT mutations expressing the p30 isoform. They concluded that loss of TET2 binding to the Gata2 distal enhancer led to increased DNA methylation in the promoter region of Gata2.

The study is interesting seeking to decipher the cooperativity between mutations in TET2 and CEBP and the following points came up regarding the mechanism.

Comments/Points:

While TET2 mutations in human AML resulted in preferential upregulation of genes (1546 up vs 1201 down), loss of TET2 in the mouse models predominantly resulted in gene downregulation (474 up vs 916 down in CEPBAp30/p30 cells; and 58 up vs 176 down in CEBPA-/p30 cells). The authors should discuss why the mouse models show this opposite gene expression pattern compared to human cells.

The authors identified 1546 up- and 1201 downregulated genes in human AML carrying combined mutations of CEBPA and TET2 compared to CEBPA-mutant patients with wild-type TET2 (Fig 1C). To complement this approach and better understand the contribution of TET2 mutations, it will be important to know how many genes are up and down in AML carrying the combined mutations of CEBPA and TET2 compared to TET2-mutant patients with wild-type CEBPA.

The authors described that the CEBPA p30 variant exerts a hypermorphic effect upregulating GATA2 expression. If so, the authors should compare and observe increased GATA2 levels in TET2 wildtype AML that carry biallelic CEBPA mutations versus TET2 wildtype AML with non-mutant CEBPA in the used datasets like Beat AML.

Is there a dose-dependent hypermorphic effect of p30 on GATA2 expression? The authors should compare and include GATA2 expression levels in CEBPA-/- vs CEBPA-/p30 and CEBPAp30/p30 myeloid cells.

Is the hypermorphic effect of p30 on GATA2 induction still observed in presence of one functional CEBPA wt allele?

The authors observed that moderate knockdown of GATA2 increased competitiveness in CEBPA p30/p30 leukemic cells (Fig 4A-D). Do the authors expect a similar competitiveness in the CEBPA-/p30 background?

It is unclear why the lower panel in Fig 5C magnifying the Gata2 enhancer shows ChIP tracks for AML-ETO. Are the ChIP tracks TET2 or AML-ETO in Fig 5C, and why are two different p30/p30 tracks? Something does not look right with the labelling.

The authors described that neutral genes displayed gene body methylation without any impact on

transcription after loss of TET2. How are neutral genes defined; and what are the expression levels? The authors should include n for the number of genes for each gene group (up, neutral, and down) in Fig 3E and Suppl Fig 3E.

In contrast to promoter DNA methylation that is associated with gene repression, gene body methylation has been associated with either no repression or transcriptional activation. Yet, the authors observed that gene body methylation was associated with downregulated genes after loss of TET2, raising the following questions:

How many of the downregulated genes that gained gene body methylation also displayed increased promoter methylation? Are there any genes that gained gene body methylation in absence of promoter hypermethylation and what is correlation with transcription in this subset of genes?

The authors should include excel tables listing all differentially expressed genes (including fold change, p values) of their results shown in Fig. 1C, Fig 2C, Fig 2G, Fig 3E.

Reviewer #4 (Remarks to the Author):

In the manuscript TET2 lesions enhance the aggressiveness of CEBPA-mutant AML by rebalancing GATA2 expression, Porse and collaborators propose a cooperative mechanism between CEBPA and TET2 mutations in AML to achieve 'just-right' levels of GATA2 which provide competitive advantage to leukaemic clones. By combining patient data analysis, and in vitro and in vivo modelling of Cebpa mutations with or without Tet2 mutation/loss with manipulation and/or reference to Gata2 expression levels, the authors propose that hypermorphic Cebpa (p30) mutations result in Gata2 overexpression by direct binding of the distal haematopoietic enhancer (DHE), resulting in minimal cell competitiveness. Tet2 mutation reduces Gata2 expression to lower-than-normal levels through promoter (Pr) methylation, and increases cell competitiveness relative to single mutants. The authors show that DNA demethylating agents can rescue the effect of Tet2 mutation in Cebpa mutant AML and restore Gata2 expression. TET2 and GATA2 mutations co-exist at lower-than-expected level in CEBPA mutant AML, supporting the notion that the two mutations work in the same pathway.

The study is systematic and generally well performed, clearly written, and with good detail in the methodological sections. It suggests a biological basis for co-existing and mutually-exclusive AML mutations, which sets a paradigm against other pairs or trios of mutational events can be assessed and potentially targeted. However, the data is to a considerable extent correlative and indicative, with no conclusive demonstration of CEBPA / TET2 interaction in the Gata2 locus, or mechanistic framework for 'just-right' GATA2 and downstream target levels.

MAJOR POINTS

1. The Authors do not show CEBPA p30 interaction with TET2 on (or off) the Gata2 locus or Gata2 DHE-Pr conformational changes upon Cebpa mutation and Tet2 mutation or loss. Ideally, the Authors would show Gata2 locus Tet2 binding with Cebpa, or dynamic promoter occupancy upon Cebpa mutation and Tet2 loss. The Authors discuss the absence of a reliable TET2 antibody citing previous work (2019) by the Helin group. However, I can only find reference in the Methods to 1 TET2 antibody and would feel reassured if a more systematic comparison, namely with (some) other antibodies expected to perform in ChIP/ChIP-seq, was included. Even if using DNA methylation as a proxy, could the authors consider performing 3C from the Gata2 DH enhancer and promoter viewpoints to complement CEBPA enhancer binding and promoter methylation changes? Functionally, the Authors could attempt to mimic gene expression regulation and functional competitive advantage of Gata2 reduction by directing a methylating tool to the Gata2 promoter in the presence of hypermorphic CEBPA p30, or conversely mutating / blocking access of CEBPA to the Gata2 DHE in the presence of Tet2mut and performing functional analyses of transformation and competition, in addition to Gata2 expression measurements. Furthermore, the Authors could attempt to reverse the effects of Tet2 mutation in lowering Gata2 expression by overexpression of a titrated form of inducible Gata2; or they could attempt to lower it further by Gata2 shRNA and verify the absence of effect (or if too low expression, putative partial reversal) of the Tet2 mutation.

2. The hypothesis of 'just-right' levels of GATA2 is an interesting one, particularly as for example

both overexpression and knockdown / HET losses of Gata2 have been shown to result in exit from the cell cycle into quiescence. The Authors present compelling data as to the competitive effects of the shRNA series in Cebpa p30 mutant cells. However, it is not clear how the levels of expression upon knockdown directly compare with those obtained upon Tet2 mut as the degree of 'functional' Gata2 KD in Fig. 4B-D seems distinct from that obtained in Fig. 3G, although not in S6A. On the functional mechanism of the 'just-right' Gata2 expression, similar cell cycle exit effects at 'extreme' GATA2 levels present a credible mechanism for a narrow window of competition advantage, which the Authors could test specifically.

MINOR POINTS

1. The Authors show the relative depletion of simultaneous TET2 and GATA2 mutations in the CEBPA mut background. Are there biological differences between triple CEBPA/TET2/GATA2 mutants and double CEBPA/TET2 or double CEBPA/GATA2 ones? Are patient survivals comparable? Are gene expression patterns comparable? Can relative competition be modelled?
2. The Authors identify 2 genes – FUT8 and SIRT5 – in addition to GATA2 which are downregulated in human and mouse CebpaMUT Tet2MUT. They do not seem to be affected at the level of promoter methylation: are they CEBPA and/or GATA2 direct targets? Do they participate in the GATA2-mediated effects?

RESPONSE TO REVIEWERS' COMMENTS

Reviewer #1 (Remarks to the Author)

The authors study cooperation between TET2 and CEBPA mutations in AML by analyzing published human patient datasets and experimental modeling in mice and with murine cells *in vitro*.

They show that synchronous deletion of Tet2 and Cebpa p42 results in an accelerated leukemia phenotype in mice as compared to either mutation alone. Comparing human datasets with *in vitro* and *in vivo* mouse data, they find that Fut8, Gata2, and Sirt5 are conserved (across species and in these different experimental setups) as targets downregulated upon loss of Tet2. Among these targets, only the Gata2 V2 promoter shows changes in promoter methylation. The authors then show that 25-60% downregulation by RNAi (but not >75% downregulation) and heterozygous deletion by CRISPR/Cas9 (but not homozygous deletion) confers a competitive advantage in Cebpa p42-deficient cells. Meanwhile, the Gata2 -77 kb enhancer is bound by Cebpa p30 and this binding is associated with increased Gata2 expression, leading the authors to conclude that Tet2 co-mutations “rebalance” Gata2 levels deranged by Cebpa mutations. Finally, azacitidine *ex vivo* restores Gata2 expression levels on this genomic background and improves survival.

Overall, the text and figures are well designed and described, and the story is clear and quite comprehensive. Some issues require further clarification.

My major comment has to do with the data flitting between somewhat different models to synthesize the results presented: When used at its best (as in Fig. 3a) it can help to narrow targets down to a small handful of conserved ones. However, in Fig. 2 (as an example), it is not clear why the authors move between Cebpa(p30/p30) cells *in vitro* and Cebpa(Δ /p30) mice [*], study Tet2(-/-) effects in the mice to model human TET2 mutations (presumably heterozygous), and compare Cas9 targeting in a myeloid progenitor cell pool *in vitro* with transformed blasts *in vivo*.

While in some respects it is not uncommon to use Tet2(-/-) mice in such studies, the discrepancy becomes somewhat more salient in this case where the finding is that Tet2 mediates a change in Gata2 expression promoting leukemogenesis with 25-60% downregulation but not with >75% downregulation.

[*] The authors call this Cebpa(-/p30), but it is an Mx1-Cre inducible deletion after transplant.

We thank the reviewer for carefully reviewing the manuscript.

The choice to use different models was mainly based on practical reasons. Specifically, we wanted to be able to use an inducible model *in vivo* and the Cebpa^{p30/p30} model was not feasible, where the use of fetal liver cells is the only possible option (see Kirstetter et al.¹ and Schuster et al.²). Moreover, the use of the Cebpa^{fl/p30} model also provided the possibility to induce both the functional Cebpa mutant and Tet2-loss at the same time point *in vivo* post-transplantation (text change made in the Materials and Methods section page 16, line 19). The fetal liver-derived Cebpa^{p30/p30} model represents a very practical *in vitro* culture system which is amenable to genetic manipulation, enabling us to test specific hypotheses. In contrast, *in vivo* derived Cebpa ^{Δ /p30} AML cells (\pm Tet2 loss) display quite limited growth in culture thereby limiting our ability to manipulate these cells. We have changed Cebpa^{-/p30}Tet2^{-/-} to Cebpa ^{Δ /p30}Tet2 ^{Δ/Δ} throughout the manuscript ($- \rightarrow \Delta$), and other corresponding inducible knockouts, to emphasize the use of an inducible system more clearly (marked with red in the text).

Regarding the difference between Cebpa^{p30/p30} and Cebpa ^{Δ /p30}, we observe very similar disease latencies in both models (see Kirstetter et al.¹ and Figure 2F) with both models developing AML within a year. Both models are characterized by a complete loss of the full-length CEBPA isoform (CEBPA p42). This appears to be a prerequisite for AML initiation since mice which retain hematopoietic expression of p42 (Cebpa^{+/p30}) very sporadically develop the disease (Supplemental figure 2E and unpublished data). Importantly, Cebpa expression levels does not differ between Cebpa^{fl/p30} and Cebpa^{fl/fl} mice. Also, Cebpa expression levels in primary leukemia does not significantly differ between Cebpa^{p30/p30} and Cebpa ^{Δ /p30}. A note of this has been added to the Results section (page 6, lines 8–11 and 13–15) and the Materials and Methods section (page 16, lines 25 and page 16, lines 1–4), and one co-author has been added (page 1, line 4).

When comparing Cebpa expression in murine primary AML blasts to established (i.e., re-transplanted) leukemia utilized in the Gata2 knockdown experiment, we observed a significant decrease in Cebpa mRNA levels, assessed by RNA sequencing, with 56–73% (Jakobsen et al.³ (Fold change 0.27; FDR 0.00093) and Trempenau et al.⁴

(Fold change 0.44; FDR 0.0029), respectively). This data has been added to the Results section (page 7, lines 37–38) and Materials and Methods section (page 19, line 5–6). The Supplemental Materials and Methods section has been updated accordingly. The differences in *Cebpa* levels, and hence the resulting changes in *Gata2* levels, in the inducible and establish AML models may likely affect the “optimal” or “required” level of *Gata2* down-regulation promoting AML. We have removed the defined percentages from the text in order not to mislead the reader.

Finally, we have used a full loss of *Tet2* in our two experimental models with CRISPR-targeting generating homozygous mutations and homozygous knockout of the *Tet2* alleles in the *in vivo* setting. We are aware that this is not in line with patient data where most patients present with mono-allelic *TET2* loss-of-function mutations (even though biallelic *TET2* mutations are also found), resulting in haploinsufficiency. However, as the reviewer correctly points out the induction of biallelic *Tet2* loss is predominantly used in murine models in the field of experimental hematology as heterozygous *Tet2* lesions display no phenotype, neither in clonal hematopoiesis (unless challenged with IL1, see Caiado et al.⁵) or in AML – perhaps due to the shorter lifespan of mice. We do agree that this is a limitation of our study, and a corresponding statement has been added to the Discussion section (page 11, lines 11–13).

Additional specific comments

1. p. 7, line 33: “concomitant mutations in both genes would be redundant and thus . . . mutually exclusive” — one would think that it is synthetic lethality that would explain mutual exclusivity; it’s not clear that this line of argumentation here is really necessary: based on observational human data, there is a slight paucity of co-occurring *Tet2* and *Gata2* mutations, but the mechanistic work isn’t being undertaken to distinguish redundancy from synthetic lethality.

We thank the reviewer for this insightful comment. Given that the two mutations would result in loss-of *GATA2* (function) and that full loss of *Gata2* results in terminal differentiation of *Cebpa*^{p30/p30} cells (Schmidt et al.⁶), synthetic lethality would also be a plausible explanation for the decreased frequency of *TET2-GATA2* double mutated AMLs. The text in the Results section has been changed accordingly (page 8, lines 21–23).

2. p. 8, line 23: In what cell type is this experiment undertaken? Neither the text nor figure legend makes this clear. Do the results differ in different cell types?

We have clarified the cell type used in this experiment in the Result section and corresponding figure legend. Information of the cell type and experimental conditions have been added to the Results section (page 9, line 9) and the legend to Figure 5D-E. Other cell types than *Cebpa*^{p30/p30} cells were not tested.

3. p. 8, lines 28-30: In this specific experiment with CRISPR/Cas9-mediated disruption of the *Gata2* enhancer element bound by *Cebpa*, is abrogation of binding to that enhancer element associated with decreased *Gata2* V2 expression (as opposed to calling back to a distinct experiment in Fig. 3F-G)?

Data in Figure 5D-E represent total *Gata2* mRNA levels, and this is now clarified in the legend. Targeting of the *G2DHE* affects expression of both *Gata2* V1 and V2, and these data have been added to Supplemental Figure 5H–K. To illustrate the changes in expression levels more clearly, the average change in *Gata2* expression upon the deletions has been added as new graphs (Figure 5E, Supplemental figure 5I+K+L).

In relation to adding *Gata2* V1 and V2 data, we have removed the data for the CRISPR/cas9 targeting of the *G2DHE* since the experiments with deletions of fragments of the *CEBPA*-bound region using two sgRNAs is more informative (Former Figure 5E has been removed as well as related text).

Additionally, to further strengthen the notion that the effect of *CEBPA* on *Gata2* expression is *TET2*-dependent, we introduced the same 250-500 bp deletions in the *CEBPA*-bound region of the *G2DHE* using CRISPR/Cas9 with two sgRNAs in *Cebpa*^{p30/p30}*Tet2*^{MUT} cells and evaluated *Gata2* expression level with qPCR. In contrast to the results of the deletions in *TET2*-proficient cells, *Gata2* levels did not change in *TET2*-deficient cells. The data has been added to Supplemental figure 5L+M.

Changes have been made to the Results section (page 9, lines 9–12), Materials and Methods section (Page 16, lines 9–12), Figure 5 and Supplemental figure 5 including legends, and Supplemental table 5.

4. p. 9, lines 12-28; Fig. 6: The relative expression of *Gata2* mRNA is decreased to nearly zero in *Tet2*-deficient mice (without azacitidine treatment): yet earlier in the work it is said that efficient downregulation of *Gata2* expression (>75%) does not provide a competitive advantage and, in fact, homozygous deletion of *Gata2* is not viable. How would the authors explain the apparent contradiction here? Is the analysis confounded by a higher percentage of dying cells, more rapid cell cycling, different cell maturation stage at this time point?

We do agree that the data on *Gata2* expression after 5-AZA treatment are not fully in line with other results of our work. The initial results in Figure 6A–C were generated with one clone of *Tet2*-proficient and *Tet2*-deficient *Cebpa*^{Δp30} AML, respectively. We acknowledge that the original *Cebpa*^{Δp30} *Tet2*^{ΔΔ} clone (A) exhibited extremely low *Gata2* levels compared to primary *Tet2*-deficient leukemic blasts (Supplemental figure 3B) and the *Gata2* shRNA-experiment (Figure 4B and Supplemental figure 4A). We have now repeated the experimental setup with one additional *Tet2*-proficient clone and two *Tet2*-deficient clones. The newly tested *Cebpa*^{Δp30} *Tet2*^{ΔΔ} clones (B–C) showed *Gata2* levels that were comparable to the other data sets. Clone A gave rise to higher blast expansion at day 16 post-transplant, indicating a high proliferation rate, which may explain the very low *Gata2* levels. Data from the additional clones tested has been added to Figure 6A and Figure 6B. Corresponding changes were introduced into the figure legend as well as the Results section (page 10, line 9 and lines 11–16)

Reviewer #2 (Remarks to the Author)

Mutational cooperativity is a critical part of leukemogenesis, and Heyes et al. sought to illuminate the mechanistic basis for the cooperation between mutations in CEBPA and TET2 in acute myeloid leukemia (AML). The CEBPA gene is biallelically mutated (double mutated; CEBPA-DM) in a subset of AML patients that harbor either biallelic CEBPA N-terminal (NT) mutations or a combination of a NT mutation together with a C-terminal (CT) mutation in the other allele. CEBPA NT lesions promote the expression of a truncated hypermorphic p30 isoform, whereas CEBPA CT mutations result in amorphic CEBPA variants unable to dimerize or bind DNA. To study the impact of double mutant CEBPA in AML, the authors used cell and murine models with a CEBPA-/p30 and CEBPA p30/p30 genetic background. The former presumably mimicking the double mutant CEBPA with a NT and CT mutation, while the latter mimicking the biallelic CEBPA NT mutations. Next, to understand the impact of double mutant CEBPA AML and co-occurring lesions with TET2, the authors introduced TET2 mutations with CRISPR or by conditional knockout of the *Tet2* alleles. The authors described that TET2 loss-of-function in double mutant CEBPA AML led to an aggressive disease phenotype by rebalancing the increased and suboptimal levels of GATA2 that are induced by hypermorphic CEBPA NT mutations expressing the p30 isoform. They concluded that loss of TET2 binding to the *Gata2* distal enhancer led to increased DNA methylation in the promoter region of *Gata2*.

The study is interesting seeking to decipher the cooperativity between mutations in TET2 and CEBP and the following points came up regarding the mechanism.

We thank the reviewer for carefully reviewing the manuscript and for the encouraging comments.

Comments/Points:

1. While TET2 mutations in human AML resulted in preferential upregulation of genes (1546 up vs 1201 down), loss of TET2 in the mouse models predominantly resulted in gene downregulation (474 up vs 916 down in CEPBAp30/p30 cells; and 58 up vs 176 down in CEBPA-/p30 cells). The authors should discuss why the mouse models show this opposite gene expression pattern compared to human cells.

The somewhat unexpected finding that more genes are up- vs. downregulated upon loss of function of TET2 in a patient setting, may be explained by the fact that most *TET2*^{MUT} AML patients have a monoallelic TET2 mutation and one remaining functional *TET2*^{WT} allele, and hence, have remaining DNA-demethylase activity from that allele. In our mouse and cell models we induce a homozygous loss of *Tet2*, which may generate a more pronounced global repression of gene expression. Furthermore, other co-occurring mutations may contribute to de-regulated transcription in human patients with *TET2* mutations. We also have experienced that, in general, it is easier to identify up- than down-regulated genes in highly heterogenous datasets, such as RNA-seq data from AML patients. We have added an explanation in the Results section (page 5, lines 15–18).

2. The authors identified 1546 up- and 1201 downregulated genes in human AML carrying combined mutations of CEBPA and TET2 compared to CEBPA-mutant patients with wild-type TET2 (Fig 1C). To complement this approach and better understand the contribution of TET2 mutations, it will be important to know how many genes

are up and down in AML carrying the combined mutations of CEBPA and TET2 compared to TET2-mutant patients with wild-type CEBPA.

To address this question, we made use of the BeatAML data set (Tyner et al.⁷; Vizome.org):

When analyzing RNA-seq data from *CEBPA*^{NT}*TET2*^{MUT} (n=5) compared to *CEBPA*^{WT}*TET2*^{MUT} (n=34) we find very few genes deregulated (14 up and 244 down). This is likely due to the heterogenous nature of the reference group, which includes many different driver mutations in addition to the *TET2* mutations. In any event, as these numbers reflect the difference between *CEBPA*^{NT} and *CEBPA*^{WT} AML, we have decided not to include them in manuscript as this is not the focus of the present work.

To complement this analysis and our previous analysis (*CEBPA*^{NT}*TET2*^{MUT} vs *CEBPA*^{NT}*TET2*^{WT}; Figure 1), and to address the effects of *TET2*^{MUT} in a *CEBPA*^{WT} background, we also analyzed *CEBPA*^{WT}*TET2*^{MUT} patients vs. *CEBPA*^{WT}*TET2*^{WT} patients. Similar to the findings presented in Figure 1, we find comparable numbers of up- and down-regulated (601 up and 527 down) demonstrating that this property is not restricted to *CEBPA*^{NT} AML. These data are described in the Results section (page 5, lines 12–13) and Materials and Methods section (page 15, lines 25–27).

3. The authors described that the CEBPA p30 variant exerts a hypermorphic effect upregulating GATA2 expression. If so, the authors should compare and observe increased GATA2 levels in TET2 wildtype AML that carry biallelic CEBPA mutations versus TET2 wildtype AML with non-mutant CEBPA in the used datasets like Beat AML.

We thank the reviewer for this suggestion. We have added data on *GATA2* and *CEBPA* expression levels in *CEBPA*^{WT} AML (normal karyotype) vs. *CEBPA*^{NT} AML (same patients as in Figure 1) without TET2 mutations (as well as IDH1/2 and WT1 mutations since these have been shown to affect TET2 binding/enzyme activity). Indeed, we find that both *CEBPA* and *GATA2* expression levels increased in *CEBPA*^{NT} vs. *CEBPA*^{WT} AML. These data have been added to Supplemental figure 5B–C and are referred to in the Results section (page 8, lines 33–36) and the Materials and Methods section (page 15, lines 27–32).

4. Is there a dose-dependent hypermorphic effect of p30 on GATA2 expression? The authors should compare and include GATA2 expression levels in CEBPA^{-/-} vs CEBPA^{-/p30} and CEBPA^{p30/p30} myeloid cells.

Cebpa^{ΔΔ} hematopoietic cells have a differentiation block at the preGM stage (Pundhir et al.⁸) and does not differentiate to GMPs and beyond. Therefore, complete loss of *Cebpa* is incompatible with the development of cells that are at a comparable state to *Cebpa*^{Δ/p30} and *Cebpa*^{p30/p30} leukemic blasts (that are mainly Mac1⁺Gr1⁺). Hence, *Cebpa*^{ΔΔ} cells have not been included in the analyses.

Consistent with the unaltered *Cebpa* expression in *Cebpa*^{Δ/p30} vs. *Cebpa*^{p30/p30} primary leukemic blasts, we did not find any changes in *Gata2* expression between *Cebpa*^{Δ/p30} vs. *Cebpa*^{p30/p30} leukemic blasts. Text changes have been made in the Results section (page 6, lines 8–11) and the Materials and Methods section (page 16, lines 25 and page 17, lines 1–4).

However, *Cebpa* levels in established (i.e., re-transplanted) *Cebpa*^{p30/p30} leukemia is increased (+56–73%) compared to primary *Cebpa*^{Δ/p30} leukemia. This is also in line with a significant increase in *Gata2* levels in the established leukemia with 45–56% compared to primary *Cebpa*^{Δ/p30} leukemia. Data sets from established *Cebpa*^{p30/p30} (Jakobsen et al.³ (Fold change 0.44; FDR 0.00029) and Trempenau et al.⁴ (Fold change 0.55; FDR 0.017), respectively). These data have been added to the Results section (page 7, lines 37–38), Materials and Methods section (page 19, lines 5–6), and Supplemental Materials and Methods.

5. Is the hypermorphic effect of p30 on GATA2 induction still observed in presence of one functional CEBPA wt allele?

In order to attempt to answer this question, we have analyzed *Cebpa* and *Gata2* gene expression with qPCR in sorted cells from the control mice included in the leukemia initiation experiment (Supplemental Figure 2E). We sorted live donor-derived non-lymphoid and non-erythroid cells (DAPI⁻CD45.2⁺CD3⁻B220⁻Ter119⁻; same gating strategy as for the leukemia blasts) as well as myeloid cells (DAPI⁻CD45.2⁺CD3⁻B220⁻Ter119⁻Mac1⁺Gr1⁺) from mice transplanted with *Cebpa*^{fl/+} and *Cebpa*^{fl/p30}, which were frozen at the one-year time point when the experiment was terminated. Neither *Cebpa* nor *Gata2* levels were altered between the two groups. A small note of these results has been added to the Result section (page 6, lines 13–15).

6. The authors observed that moderate knockdown of GATA2 increased competitiveness in CEBPA p30/p30 leukemic cells (Fig 4A-D). Do the authors expect a similar competitiveness in the CEBPA-/p30 background?

Since *Gata2* levels are increased in the established *Cebpa*^{p30/p30} compared to primary *Cebpa*^{Δ/p30} leukemia, we would expect that the window for knockdown of *Gata2* may be narrower if we had used a primary leukemia compared to an established (i.e., re-transplanted) leukemia.

7. It is unclear why the lower panel in Fig 5C magnifying the *Gata2* enhancer shows ChIP tracks for AML-ETO. Are the ChIP tracks TET2 or AML-ETO in Fig 5C, and why are two different p30/p30 tracks? Something does not look right with the labelling.

Figure 5C has been revised to better annotate the ChIP-seq tracks included. Further, we have now replaced the TET2 ChIP-seq tracks from AML/ETO expressing cells (Rasmussen et al⁹), to include our own TET2 ChIP-seq from *Cebpa*^{p30/p30} cells *in vitro*. Corresponding changes have been made to Figure 3H.

8. The authors described that neutral genes displayed gene body methylation without any impact on transcription after loss of TET2. How are neutral genes defined; and what are the expression levels? The authors should include n for the number of genes for each gene group (up, neutral, and down) in Fig 3E and Suppl Fig 3E.

We had defined neutral genes as protein coding genes which are not transcriptionally deregulated between *Cebpa*^{Δ/p30} vs. *Cebpa*^{Δ/p30}*Tet*^{Δ/Δ} blasts. Thus, the neutral genes included both expressed and unaltered genes as well as genes that are not expressed. To highlight the impact on expressed genes, we have now stratified the data to separate expressed genes that are not differently expressed (DE) vs. not expressed genes. New graphs with these four groups have been included in Figure 3E and Supplemental figure 3E. The number of genes in each group has been added to the figure legends. Corresponding changes to the text in the Results section have been included (Page 7, lines 11–13).

The expression levels reported as median[25th-75th percentile] were as follows:

	Cebpa ^{Δ/p30} Tet2 ^{+/+}	Cebpa ^{Δ/p30} Tet2 ^{Δ/Δ}
Up (n=58)	359[44–1093]	1029[151–2118]
Not DE (n=14816)	320[29–1010]	326[24–1115]
Not expressed (n=6996)	-	-
Down (n=176)	232[91–598]	31[6–136]

9. In contrast to promoter DNA methylation that is associated with gene repression, gene body methylation has been associated with either no repression or transcriptional activation. Yet, the authors observed that gene body methylation was associated with downregulated genes after loss of TET2, raising the following questions: How many of the downregulated genes that gained gene body methylation also displayed increased promoter methylation? Are there any genes that gained gene body methylation in absence of promoter hypermethylation and what is correlation with transcription in this subset of genes?

We observed a general increase in gene body DNA methylation upon *Tet2*-deficiency with +10.0%; Δ6.38, and also down-regulated genes exhibited increased gene body methylation (+9.5%; Δ6.42). However, 35% of the down-regulated genes which gained gene body methylation upon loss of *Tet2*, also exhibited promoter hypermethylation, compared to 18% of the remaining genes (p=0.0427). The majority of genes that gained gene body methylation in absence of promoter hypermethylation, belongs to neutral genes (i.e., the genes that are not differentially expressed or not expressed; 99%). But, if the numbers of genes in each group are weighted (down/neutral/up) there is a trend towards increased frequency of genes with gain in gene body methylation within the up-regulated genes compared to neutral and down-regulated genes (14% vs 7%; p=0.0518). These data have been added to the Result section (page 7, lines 14–19).

10. The authors should include excel tables listing all differentially expressed genes (including fold change, p values) of their results shown in Fig. 1C, Fig 2C, Fig 2G, Fig 3E.

A supplemental Excel file has been added including the differentially expressed genes from *CEBPA*^{NT} vs. *CEBPA*^{NT}*TET2*^{MUT} (Sheet1), *CEBPA*^{WT} vs. *CEBPA*^{WT}*TET2*^{MUT} (Sheet2), *Cebpa*^{p30/p30}*Tet2*^{WT} vs. *Cebpa*^{p30/p30}*Tet2*^{MUT} cells (Sheet3), and *Cebpa*^{Δ/p30}*Tet2*^{+/+} vs. *Cebpa*^{Δ/p30}*Tet2*^{Δ/Δ} blasts (Sheet4), as well as promoter

methylation levels for the differentially expressed genes from *Cebpa*^{Δp30}*Tet2*^{+/+} vs. *Cebpa*^{Δp30}*Tet2*^{Δ/Δ} blasts (Sheet5).

Reviewer #4 (Remarks to the Author):

In the manuscript TET2 lesions enhance the aggressiveness of CEBPA-mutant AML by rebalancing GATA2 expression, Porse and collaborators propose a cooperative mechanism between CEBPA and TET2 mutations in AML to achieve ‘just-right’ levels of GATA2 which provide competitive advantage to leukaemic clones. By combining patient data analysis, and in vitro and in vivo modelling of *Cebpa* mutations with or without *Tet2* mutation/loss with manipulation and/or reference to *Gata2* expression levels, the authors propose that hypermorphic *Cebpa* (p30) mutations result in *Gata2* overexpression by direct binding of the distal haematopoietic enhancer (DHE), resulting in minimal cell competitiveness. *Tet2* mutation reduces *Gata2* expression to lower-than-normal levels through promoter (Pr) methylation, and increases cell competitiveness relative to single mutants. The authors show that DNA demethylating agents can rescue the effect of *Tet2* mutation in *Cebpa* mutant AML and restore *Gata2* expression. TET2 and GATA2 mutations co-exist at lower-than-expected level in CEBPA mutant AML, supporting the notion that the two mutations work in the same pathway.

The study is systematic and generally well performed, clearly written, and with good detail in the methodological sections. It suggests a biological basis for co-existing and mutually-exclusive AML mutations, which sets a paradigm against other pairs or trios of mutational events can be assessed and potentially targeted. However, the data is to a considerable extent correlative and indicative, with no conclusive demonstration of CEBPA / TET2 interaction in the *Gata2* locus, or mechanistic framework for ‘just-right’ GATA2 and downstream target levels.

We thank the reviewer for the careful review of our manuscript.

MAJOR POINTS

1. The Authors do not show CEBPA p30 interaction with TET2 on (or off) the *Gata2* locus or *Gata2* DHE-Pr conformational changes upon *Cebpa* mutation and *Tet2* mutation or loss. Ideally, the Authors would show *Gata2* locus *Tet2* binding with *Cebpa*, or dynamic promoter occupancy upon *Cebpa* mutation and *Tet2* loss. The Authors discuss the absence of a reliable TET2 antibody citing previous work (2019) by the Helin group. However, I can only find reference in the Methods to 1 TET2 antibody and would feel reassured if a more systematic comparison, namely with (some) other antibodies expected to perform in ChIP/ChIP-seq, was included. Even if using DNA methylation as a proxy, could the authors consider performing 3C from the *Gata2* DH enhancer and promoter viewpoints to complement CEBPA enhancer binding and promoter methylation changes? Functionally, the Authors could attempt to mimic gene expression regulation and functional competitive advantage of *Gata2* reduction by directing a methylating tool to the *Gata2* promoter in the presence of hypermorphic CEBPA p30, or conversely mutating / blocking access of CEBPA to the *Gata2* DHE in the presence of *Tet2*mut and performing functional analyses of transformation and competition, in addition to *Gata2* expression measurements. Furthermore, the Authors could attempt to reverse the effects of *Tet2* mutation in lowering *Gata2* expression by overexpression of a titrated form of inducible *Gata2*; or they could attempt to lower it further by *Gata2* shRNA and verify the absence of effect (or if too low expression, putative partial reversal) of the *Tet2* mutation.

We thank the reviewer for raising this point which we are well aware of. Throughout the period from submission to revision, we have tested additional TET2-antibodies to be able to verify TET2 binding to the *G2DHE* with ChIP-qPCR. Gratifyingly, we identified one antibody that works in CHIP-experiments. Using this tool, we were able to verify that TET2-binding to the *G2DHE* was decreased upon shRNA knockdown of *Cebpa* in *Cebpa*^{p30/p30} cells. Conversely, loss of TET2 did not affect CEBPA binding to the *G2DHE* (Supplemental figure 5G). We think that this finding clearly demonstrates the involvement of CEBPA in recruiting TET2 to the *G2DHE* thereby significantly strengthening the mechanistic aspects of our work. These data have been added to Figure 5I-J, and text added to the Results section (page 9, lines 24–26) and Materials and Methods section (page 16, lines 13–17; page 18, lines 20–24) as well as Supplemental Materials and Methods. The Discussion has been revised to reflect these new data (page 12, lines 16–17+19+27–30)

In addition, to further strengthen the notion that the effects of CEBPA on *Gata2* expression is TET2-dependent, we introduced 250-500 bp deletions in the CEBPA-bound regions of the *G2DHE* using CRISPR/Cas9 with the two sgRNA approach in *Cebpa*^{p30/p30}*Tet2*^{MUT} cells and evaluated *Gata2* expression. In contrast to the results of the same deletions in TET2-proficient cells, *Gata2* levels did not change in TET2-deficient cells. The data has

been added to the Supplemental figure 5L+M and corresponding text in the Results section (page 9, lines 11–12) and Materials and Methods section (page 16, lines 9–12).

We believe that the combined results of the new experiments do strengthen our mechanistic data substantially.

2. The hypothesis of ‘just-right’ levels of GATA2 is an interesting one, particularly as for example both overexpression and knockdown / HET losses of Gata2 have been shown to result in exit from the cell cycle into quiescence. The Authors present compelling data as to the competitive effects of the shRNA series in Cebpa p30 mutant cells. However, it is not clear how the levels of expression upon knockdown directly compare with those obtained upon Tet2 mut as the degree of ‘functional’ Gata2 KD in Fig. 4B-D seems distinct from that obtained in Fig. 3G, although not in S6A. On the functional mechanism of the ‘just-right’ Gata2 expression, similar cell cycle exit effects at ‘extreme’ GATA2 levels present a credible mechanism for a narrow window of competition advantage, which the Authors could test specifically.

We agree that the degree of *Gata2* reduction fluctuates between experiments. We have attributed this to a difference in basal *Gata2* expression between different leukemic clones, and since gene expression differences based on both RNA-seq and qPCR is calculated relative to a control samples, this would result in varying degrees of reduction upon KD or *Tet2*-loss. In addition, the “optimal” *Gata2* level might depend on the expression of other genes which might be deregulated upon loss of TET2. Therefore, we have removed the knockdown efficiency levels in the description of the results related to Figure 4B-D, as they might be misleading.

We have attempted to address the effects of *Gata2* knockdown on cell cycle, and to this end we evaluated the expression of *Ki67* in cells from mice transplanted with shControl vs. shGata2 A (moderate knockdown) and shGata2 D (high knockdown). Supporting the reviewer’s point of a role for GATA2 levels in cell cycle control, we see a significant increase in *Ki67* expression in the ShGata2 A, but not in the ShGata2 D group. These data have been added to Supplemental figure 4C and are described in the Result section (page 8, lines 6–8), and Materials and Methods section (page 18, line 3).

MINOR POINTS

1. The Authors show the relative depletion of simultaneous TET2 and GATA2 mutations in the CEBPA mut background. Are there biological differences between triple CEBPA/TET2/GATA2 mutants and double CEBPA/TET2 or double CEBPA/GATA2 ones? Are patient survivals comparable? Are gene expression patterns comparable? Can relative competition be modelled?

Unfortunately, RNA expression and survival data are not available from patients in the cohort studies included in the mutational status analysis. In the clinical dataset from the MLL Münchner Leukämie Labor GmbH, we find that survival of $CEBPA^{DM}TET2^{MUT}GATA2^{MUT}$ is comparable to $CEBPA^{DM}TET2^{MUT}$ patients (See Figure below). However, the statistical power of the analysis is limited by the low numbers of patients in the triple mutant cohort (n=6). Therefore, the requested data have not been included in the manuscript.

Overall survival (days) of $CEBPA^{DM}$ patients.

$GATA2^{WT}TET2^{WT}$ n=56, $GATA2^{MUT}TET2^{WT}$ n=17, $GATA2^{WT}TET2^{MUT}$ n=26, $GATA2^{MUT}TET2^{MUT}$ n=6

We have not attempted to perform a competition assay using *Gata2* KD in *Tet2*-knockout cells. We expect to lower *Gata2* levels in *Tet2*-knockout cells too strongly, and in this regard the *Gata2* levels would not be

comparable to the *Tet2*-WT setting. We consider that the results from said experiment would be complicated to interpret since, the window of “optimal” *Gata2* levels may be missed.

2. The Authors identify 2 genes – *FUT8* and *SIRT5* – in addition to *GATA2* which are downregulated in human and mouse *Cebpa*MUT *Tet2*MUT. They do not seem to be affected at the level of promoter methylation: are they *CEBPA* and/or *GATA2* direct targets? Do they participate in the *GATA2*-mediated effects?

To address this question, we have analyzed both input and output cells from the *in vivo* *shGata2* experiment (Figure 4A-D). We did not observe any change in the expression of *Fut8* nor *Sirt5* upon *Gata2*-knockdown in *Cebpa*^{p30/p30} leukemic cells. Hence, we do not have any indications that down-regulated expression of these genes participates in the *Gata2*-mediated effect on *Cebpa*^{p30/p30} leukemia.

However, *Fut8* seems to be regulated by *Cebpa* levels and/or mutational status. Specifically, *Fut8* expression is upregulated in murine *Cebpa*^{p30/p30} leukemic cells and *CEBPA*^{DM} AML compared to their normal counterpart (data from Jakobsen et al.³) and down-regulated upon *Cebpa*-knockdown in *Cebpa*^{p30/p30} cells *in vitro* (data from Heyes et al.¹⁰). *Sirt5* is only found up-regulated in murine *Cebpa*^{p30/p30} compared to normal cells, while unchanged in the human setting as well as upon *Cebpa*-knockdown (data from Jakobsen et al.³ and Heyes et al.¹⁰), thereby indicating that *Fut8* is a *CEBPA* target gene, while *Sirt5* is not.

References:

- 1 Kirstetter, P. *et al.* Modeling of C/EBPalpha mutant acute myeloid leukemia reveals a common expression signature of committed myeloid leukemia-initiating cells. *Cancer Cell* **13**, 299-310, doi:10.1016/j.ccr.2008.02.008 (2008).
- 2 Schuster, M. B. *et al.* Lack of the p42 form of C/EBPalpha leads to spontaneous immortalization and lineage infidelity of committed myeloid progenitors. *Exp Hematol* **41**, 882-893 e816, doi:10.1016/j.exphem.2013.06.003 (2013).
- 3 Jakobsen, J. S. *et al.* Mutant CEBPA directly drives the expression of the targetable tumor-promoting factor CD73 in AML. *Sci Adv* **5**, eaaw4304, doi:10.1126/sciadv.aaw4304 (2019).
- 4 Trempenau, M. L. *et al.* The histone demethylase KDM5C functions as a tumor suppressor in AML by repression of bivalently marked immature genes. *Leukemia* **37**, 593-605, doi:10.1038/s41375-023-01810-6 (2023).
- 5 Caiado, F. *et al.* Aging drives Tet2^{+/-} clonal hematopoiesis via IL-1 signaling. *Blood* **141**, 886-903, doi:10.1182/blood.2022016835 (2023).
- 6 Schmidt, L. *et al.* CEBPA-mutated leukemia is sensitive to genetic and pharmacological targeting of the MLL1 complex. *Leukemia* **33**, 1608-1619, doi:10.1038/s41375-019-0382-3 (2019).
- 7 Tyner, J. W. *et al.* Functional genomic landscape of acute myeloid leukaemia. *Nature* **562**, 526-531, doi:10.1038/s41586-018-0623-z (2018).
- 8 Pundhir, S. *et al.* Enhancer and Transcription Factor Dynamics during Myeloid Differentiation Reveal an Early Differentiation Block in *Cebpa* null Progenitors. *Cell Rep* **23**, 2744-2757, doi:10.1016/j.celrep.2018.05.012 (2018).
- 9 Rasmussen, K. D. *et al.* TET2 binding to enhancers facilitates transcription factor recruitment in hematopoietic cells. *Genome Res* **29**, 564-575, doi:10.1101/gr.239277.118 (2019).
- 10 Heyes, E. *et al.* Identification of gene targets of mutant C/EBPalpha reveals a critical role for MSI2 in CEBPA-mutated AML. *Leukemia* **35**, 2526-2538, doi:10.1038/s41375-021-01169-6 (2021).

The following changes has been made to the manuscript to meet Nature Communications policies:

- 1) RRIDs has been added when applicable.
- 2) Information regarding sex of patients, mice, and cell lines has been added when missing.
- 3) Gating strategy for flow cytometry have been added (Supplemental Figure 2E).
- 4) Uncropped blots are provided in the supplemental excel file.
- 5) Graphs has been changed to dot plots to visualize individual data points.
- 6) An ethics statement has been added to the methods section (page 14, lines 15-18).
- 7) Affiliation #2 has been updated to the current version for the Copenhagen University Hospital (page 1, line 9).
Affiliation #7 has been added for co-author FG (page 1, line 15).
Affiliation #8 has been added for co-author BTP (page 1, line 16).
- 8) Parts of the Materials and Methods section have been moved to the Supplemental Materials and Methods to better meet the word limits.

All changes to the text in the manuscript are marked in **Red**.

REVIEWER COMMENTS

Reviewer #1 (Remarks to the Author):

Responses to reviewers and corresponding edits do address specific concerns outlined in the initial review comments. The revisions particularly to Fig. 5 (in response to multiple reviewers) strengthen the work and it is commendable that authors were able to find a working Tet2 antibody for ChIP.

Regarding the major comment about varying mice and murine cells used for mechanistic work, authors' response regarding practical limitations is well taken. Based on the revised data noted on page 6, lines 8-11 and 13-15, there is a (reasonable but significant) limitation of the study which is important to call out and I would suggest should be noted at least in the Discussion:

Association with relatively 'favorable prognosis' in human CEBPA-mutated AML is now attributed to the presence of a C-terminal basic leucine zipper (bZIP) in-frame mutation (Tarlock et al, Blood (2021); Wakita et al, Blood Adv (2022); and others); indeed recent revisions to clinical prognostic criteria recognize CEBPA(bZIP-inf) AML specifically as the only CEBPA-mutated entity with favorable prognosis.

In this revision, authors here reveal that their preferred in vivo model of Cebpa-mutant AML (Δ /p30) has similar disease latencies and Cebpa gene expression as their preferred in vitro model (p30/p30; elsewhere in the literature known as L/L). By contrast, the published mouse model (K/L) of biallelic Cebpa AML with C-terminal in-frame mutation has different (shorter) disease latency than does p30/p30 (L/L) (Bereshchenko et al., Cancer Cell (2009)). Models used in this paper, therefore, have both mutational configurations and disease kinetics consistent with non-'favorable prognosis' AML rather than CEBPA(bZIP-inf) AML, the clinical entity described by authors in their Introduction that would be expected to predominate in the human cohort described in Fig. 1.

This does not refute the possibility that cooperation between Cebpa and Tet2 at the Gata2 locus as described by the authors in their mechanistic work may also be at play in explaining poorer outcomes shown in Fig. 1B when patients have both TET2 and CEBPA mutations, but conversely it does not seem appropriate to generalize from discrepant mouse models to the human disease entity without comment. It'd be unreasonable to expect authors to produce additional experiments to address this point, but this major limitation inherent to the tools at their disposal should be made clear in the Discussion.

Reviewer #2 (Remarks to the Author):

In this revised manuscript, the authors have strengthened their paper and addressed my previous points/questions.

Regarding the new data, the following points/questions came up:

Figure 5J shows TET2 binding at G2DE assessed by ChIP-qPCR. Are these cells still healthy after knockdown of CEBPA and do TET2 levels remain stable following CEBPA knockdown? This is important to exclude the possibility that reduced TET2 binding results from lower intracellular TET2 levels after knockdown of CEBPA.

The fixation for ChIP with 11% formaldehyde seems excessive compared to the usual 1% formaldehyde and raises concerns about crosslinking artefacts. Since TET2-ChIP is challenging using non-tagged approaches, the authors may consider as a control using TET2 knockdown approaches to test that the ChIP-qPCR can't detect TET2 enrichment at GDH2 after CEPBA knockdown.

Related to the point above, the ChIP-seq tracks for TET2 on Fig 3H show rather a random distribution pattern along the track without clear enrichment, especially considering that the peaks vary only in a range of 0 - 0.27. This range is typically observed as a background signal. The

authors may consider IP approaches for 5hmC as a surrogate and validation of TET2 binding sites.

Reviewer #4 (Remarks to the Author):

I thank the Authors for the additional experiments and clarification of the points raised. I am satisfied with the changes to the manuscript in response to my comments.

RESPONSE TO REVIEWERS' COMMENTS

Reviewer #1 (Remarks to the Author)

Responses to reviewers and corresponding edits do address specific concerns outlined in the initial review comments. The revisions particularly to Fig. 5 (in response to multiple reviewers) strengthen the work and it is commendable that authors were able to find a working Tet2 antibody for ChIP. Regarding the major comment about varying mice and murine cells used for mechanistic work, authors' response regarding practical limitations is well taken. Based on the revised data noted on page 6, lines 8-11 and 13-15, there is a (reasonable but significant) limitation of the study which is important to call out and I would suggest should be noted at least in the Discussion:

Association with relatively 'favorable prognosis' in human CEBPA-mutated AML is now attributed to the presence of a C-terminal basic leucine zipper (bZIP) in-frame mutation (Tarlock et al, Blood (2021); Wakita et al, Blood Adv (2022); and others); indeed recent revisions to clinical prognostic criteria recognize CEBPA(bZIP-inf) AML specifically as the only CEBPA-mutated entity with favorable prognosis.

In this revision, authors here reveal that their preferred in vivo model of Cebpa-mutant AML (Δ /p30) has similar disease latencies and Cebpa gene expression as their preferred in vitro model (p30/p30; elsewhere in the literature known as L/L). By contrast, the published mouse model (K/L) of biallelic Cebpa AML with C-terminal in-frame mutation has different (shorter) disease latency than does p30/p30 (L/L) (Bereshchenko et al., Cancer Cell (2009)). Models used in this paper, therefore, have both mutational configurations and disease kinetics consistent with non-'favorable prognosis' AML rather than CEBPA(bZIP-inf) AML, the clinical entity described by authors in their Introduction that would be expected to predominate in the human cohort described in Fig. 1.

This does not refute the possibility that cooperation between Cebpa and Tet2 at the Gata2 locus as described by the authors in their mechanistic work may also be at play in explaining poorer outcomes shown in Fig. 1B when patients have both TET2 and CEBPA mutations, but conversely it does not seem appropriate to generalize from discrepant mouse models to the human disease entity without comment. It'd be unreasonable to expect authors to produce additional experiments to address this point, but this major limitation inherent to the tools at their disposal should be made clear in the Discussion.

We thank the reviewer for the comments and are sorry not to have addressed the concerns in our previously revised manuscript. In the following, we will for simplicity use the following terms to describe the *CEBPA* genotypes: L = p30/*CEBPA*-TAD; K = C-terminal lesions/*CEBPA*(bZIP-inf). For reference see Taube et al.¹

The mouse models used in our study are either L/delta or L/L, which we deem equivalent in molecular terms as they both only express the p30 version of CEBPA. This is supported by our experiments and was addressed in our first rebuttal letter. We recognize that the L/L combination is underrepresented in *CEBPA*-mutated human AML and to our knowledge the prognostic impact of this group (given their low frequency in patients) has not been analyzed to date. Since mice with the K/L combination develop leukemia faster than animals with the L/L combination (equivalent to our models), we fail to see why our models should represent the "non favorable prognosis" AML subgroup. In support of our models being highly equivalent to human *CEBPA*^{DM} AML (i.e., largely K/L) and different from L/+ human AML we would like to highlight the following:

- 1) Compared to normal myeloid progenitors (GMPs) *Cebpa/CEBPA* is upregulated transcriptionally in leukemic GMPs (L-GMPs) both in our models and in human *CEBPA*^{DM} AML (Jakobsen et al.²). This upregulation is not seen in non-leukemic murine L/+ progenitors (see previous rebuttal letter).
- 2) Similarly, compared to normal GMPs, *Gata2/GATA2* is upregulated transcriptionally in L-GMPs both in our models and in human *CEBPA*^{DM} AML (Figure 5A, Supplemental Figure 5A and Adamo et al.³). Again, this upregulation is not seen in non-leukemic murine L/+ progenitors (see previous rebuttal letter).
- 3) Loss of TET2 accelerates leukemic development in L/delta mice (Figure 2F) and is associated with poor outcome in *CEBPA*^{DM} AML (Figure 1B and Taube et al.¹). Loss of TET2 does not impact on outcome in human L/+ AML (Taube et al.¹).

4) *Gata2/GATA2* is down-regulated in *TET2* mutated L/L cells and human *CEBPA*^{DM} AML compared to their respective *TET2*-proficient counterparts (Figure 1C and 2G).

So, in terms of the CEBPA-TET2-GATA2 axis, our L/L and L/delta models display near-identical phenotypes as those observed in human *CEBPA*^{DM} AML.

We would also like to note that the K/+ combination in mice has not been reported to result in leukemia development. Instead, K/K mice develop AML, but the resulting leukemias are mainly of erythroid origin, suggesting that this animal model does not ideally mimic human single mutated CEBPA(bZIP-inf). To us a comparison between the three main human subtypes (K/L; K/+ and N/+) represents a major experimental question in the field that will likely involve experiments in a humanized setting.

While we are confident that the L/L and L/delta combinations in mice are strongly resembling human *CEBPA*^{DM} AML (i.e., K/L) especially with regards to the CEBPA-TET2-GATA2 axis (i.e., the topic of this work), we do agree with the reviewer that we should note this in our manuscript. We have therefore added a paragraph to the discussion (page 13; lines 5–12), which we hope addresses the reviewer's concern.

Reviewer #2 (Remarks to the Author)

In this revised manuscript, the authors have strengthened their paper and addressed my previous points/questions.

Regarding the new data, the following points/questions came up:

Figure 5J shows TET2 binding at G2DE assessed by ChIP-qPCR. Are these cells still healthy after knockdown of CEBPA and do TET2 levels remain stable following CEBPA knockdown? This is important to exclude the possibility that reduced TET2 binding results from lower intracellular TET2 levels after knockdown of CEBPA.

The fixation for ChIP with 11% formaldehyde seems excessive compared to the usual 1% formaldehyde and raises concerns about crosslinking artefacts. Since TET2-ChIP is challenging using non-tagged approaches, the authors may consider as a control using TET2 knockdown approaches to test that the ChIP-qPCR can't detect TET2 enrichment at GDH2 after CEPBA knockdown.

Related to the point above, the ChIP-seq tracks for TET2 on Fig 3H show rather a random distribution pattern along the track without clear enrichment, especially considering that the peaks vary only in a range of 0 - 0.27. This range is typically observed as a background signal. The authors may consider IP approaches for 5hmC as a surrogate and validation of TET2 binding sites.

We thank the reviewer for the thorough second review of our manuscript.

The *Cebpa*^{p30/p30} cells are still healthy at the 48 hrs timepoint after induction of the shRNA (see Heyes et al.⁴ for more information about this model).

Tet2 mRNA expression, assessed by RNA-seq, is slightly reduced at this timepoint (-12%, p=0.0319), while TET2 protein levels, assessed by western blot (Rabbit anti-TET2 Antibody, #304-247A, Bethyl Laboratories), are unaltered or slightly increased. Thus, we believe that we can exclude the possibility that the reduced TET2 binding to the *G2DHE* is due to lower cellular TET2 levels.

Regrettably, the 11% formaldehyde crosslinking step in the Material and Methods section was erroneously described and we appreciate that the reviewer spotted this shortcoming. We do use an 11% formaldehyde stock solution, but the final concentration of formaldehyde is 1%. This has now been corrected (page 18; line 22–23).

To address potential non-specific binding of the TET2 antibody, we have assessed TET2 binding at the *G2DHE* by ChIP-qPCR in *Cebpa*^{p30/p30}*Tet2*^{WT} cells and two *Cebpa*^{p30/p30}*Tet2*^{MUT} clones. We find that the TET2 binding is not enriched at the *G2DHE* over a negative control region in the *Tet2*^{MUT} clones, demonstrating that the TET2 signal that we detect in the *Tet2*^{WT} cells is specific.

We agree with the conclusion that the TET2 ChIP-seq track in Figure 3H (covering ONLY the *Gata2* gene) is showing background binding – we have not claimed otherwise. In contrast, we found a distinct and specific binding of TET2 at the *G2DHE* and the difference in TET2 binding to the enhancer vs the *Gata2* promoter which is now clearly highlighted in Figure 5C. This is in agreement with previously published TET2 ChIP-seq data from Rasmussen et al.⁵, where the TET2 binding is high at the *G2DHE* but low or absent at the *Gata2* locus including its two promoter regions. We have removed the CEBPA and TET2 tracks from Figure 3H since the information regarding their binding at the *Gata2* locus is relevant for the content in Figure 5 but not for the results presented in Figure 3 and the respective text sections. Text changes have been made in the Results section and figure legends (page 7; lines 29-30, page 9; lines 4-5, page 28; line 15; and page 30; lines 7-8).

Further, we have changed the TET2 tracks in Figure 5C to show RPKM instead of CPM since we acknowledge that this difference in normalization and hence, scaling, might be confusing. All tracks are now shown as RPKM. The figure legend on page 29-30 has been changed accordingly.

We hope these control experiments and changes to the manuscript satisfy the reviewers concerns.

Reviewer #4 (Remarks to the Author)

I thank the Authors for the additional experiments and clarification of the points raised. I am satisfied with the changes to the manuscript in response to my comments.

We are pleased that the reviewer found our revisions satisfactory.

References:

- 1 Taube, F. *et al.* CEBPA mutations in 4708 patients with acute myeloid leukemia: differential impact of bZIP and TAD mutations on outcome. *Blood* **139**, 87-103, doi:10.1182/blood.2020009680 (2022).

- 2 Jakobsen, J. S. *et al.* Mutant CEBPA directly drives the expression of the targetable tumor-promoting factor CD73 in AML. *Sci Adv* **5**, eaaw4304, doi:10.1126/sciadv.aaw4304 (2019).
- 3 Adamo, A. *et al.* Identification and interrogation of the gene regulatory network of CEBPA-double mutant acute myeloid leukemia. *Leukemia* **37**, 102-112, doi:10.1038/s41375-022-01744-5 (2023).
- 4 Heyes, E. *et al.* Identification of gene targets of mutant C/EBPalpha reveals a critical role for MSI2 in CEBPA-mutated AML. *Leukemia* **35**, 2526-2538, doi:10.1038/s41375-021-01169-6 (2021).
- 5 Rasmussen, K. D. *et al.* TET2 binding to enhancers facilitates transcription factor recruitment in hematopoietic cells. *Genome Res* **29**, 564-575, doi:10.1101/gr.239277.118 (2019).

REVIEWERS' COMMENTS

Reviewer #2 (Remarks to the Author):

The authors have satisfactorily addressed my questions and comments in their response letter. I have no additional requests for revisions.

Point-to-point response (2nd revision):

Reviewer #2 (Remarks to the Author)

The authors have satisfactorily addressed my questions and comments in their response letter. I have no additional requests for revisions.

We are pleased that the reviewer found this revision satisfactory.